# FedLWS: Federated Learning with Adaptive Layer-wise Weight Shrinking

**Changlong Shi[1], Jinmeng Li[1], He Zhao[2], Dandan Guo[1*], Yi Chang[1 3 4*]**

School of Artificial Intelligence, Jilin University[1]

CSIRO's Data61[2] International Center of Future Science, Jilin University[3]

Engineering Research Center of Knowledge-Driven Human-Machine Intelligence, MOE, China[4]

`{shicl22,lijm9921}@mails.jlu.edu.cn,`

`he.zhao@data61.csiro.au, {guodandan,yichang}@jlu.edu.cn`

## Abstract

In Federated Learning (FL), weighted aggregation of local models is conducted to generate a new global model, and the aggregation weights are typically normalized to 1. A recent study identifies the global weight shrinking effect in FL, indicating an enhancement in the global model's generalization when the sum of weights (i.e., the shrinking factor) is smaller than 1, where how to learn the shrinking factor becomes crucial. However, principled approaches to this solution have not been carefully studied from the adequate consideration of privacy concerns and layer-wise distinctions. To this end, we propose a novel model aggregation strategy, Federated Learning with Adaptive Layer-wise Weight Shrinking (FedLWS), which adaptively designs the shrinking factor in a layer-wise manner and avoids optimizing the shrinking factors on a proxy dataset. We initially explored the factors affecting the shrinking factor during the training process. Then we calculate the layer-wise shrinking factors by considering the distinctions among each layer of the global model. FedLWS can be easily incorporated with various existing methods due to its flexibility. Extensive experiments under diverse scenarios demonstrate the superiority of our method over several state-of-the-art approaches, providing a promising tool for enhancing the global model in FL.

## 1 Introduction

Federated Learning (FL) as an innovative paradigm in machine learning has attracted substantial attention in recent years (Li et al., 2020a; Zhang et al., 2021; Kairouz et al., 2021; Qi et al., 2023). This distributed optimization approach allows model updates to be computed locally and aggregated without exposing raw data, thereby effectively addressing the challenges arising from distributed data sources while simultaneously preserving privacy and security. In FL, weighted aggregation of local models is conducted to generate the global model, where how to design the aggregation scheme is a critical problem (Kairouz et al., 2021; Qi et al., 2023; Ye et al., 2023).

Most previous works (Li et al., 2020b; McMahan et al., 2017) conduct model aggregation simply based on the local dataset relative size, which however could be sub-optimal empirically due to the data heterogeneity. Consequently, many methods (Ye et al., 2023; Hsu et al., 2019; Wang et al., 2020b) focus on adjusting the model aggregation scheme to enhance the global model's performance, with the majority of these methods normalizing the aggregation weights (i.e., the sum of aggregation weights is 1, represented as $\gamma = 1$, where $\gamma$ is the sum of weights). Recently, Li et al. (2023a) revisited the weighted aggregation process and gained new insights into the training dynamics of FL, identifying the Global Weight Shrinking effect (GWS, analogous to weight decay) in FL, when $\gamma$ is smaller than 1, which can further enhance the global model's generalization. The study demonstrates that during the training process, the shrinking factor $\gamma$ plays a crucial role in maintaining the balance between the regularization term and the optimization term. Li et al. (2023a) proposed FedLAW, which optimizes the value of $\gamma$ using gradient descent on a proxy dataset that has the same distribution as the global dataset. Despite its notable performance improvements, two key issues limit its practical applicability and adaptability.

---

*Corresponding authors.

Firstly, the use of proxy datasets may raise privacy-related concerns, which are particularly crucial in the context of FL. Additionally, obtaining a dataset with a distribution identical to the global dataset is challenging in practice, and discrepancies between the proxy and global data distributions can negatively impact the method's effectiveness. Secondly, divergence across different layers of deep neural networks has been demonstrated in numerous previous studies (Ma et al., 2022; Rehman et al., 2023; Lee et al., 2023). Hence, simply applying a single shrinking factor $\gamma$ across the entire model may not effectively harness the benefits of global weight shrinking."

To address the aforementioned issues, we initially conducted a series of experiments to explore factors influencing the shrinking factor's value during Federated Learning training. Building on these empirical observations from our experiments, we subsequently propose our method, Federated Learning with Adaptive Layer-wise Weight Shrinking (FedLWS). By analyzing the relationship between the regularization and optimization terms, and examining how their ratio correlates with the variance of local gradients, we derive an expression for the shrinking factor $\gamma$. This factor is computed directly using the gradients and parameters of the global model, which are readily accessible in Federated Learning. Consequently, FedLWS eliminates the need to optimize the shrinking factor $\gamma$ on a proxy dataset, as required by previous work (Li et al., 2023a). This not only addresses privacy-related concerns but also enhances its feasibility and applicability in practical, real-world deployments. Furthermore, by leveraging the gradients and parameters of each layer in the global model, it is easy to calculate the layer-wise $\gamma$ for each layer. This allows for an improvement in model generalization by considering layer-wise differences of shrinking factor. Moreover, our approach is conducted after the server-side model aggregation. That is to say, it is orthogonal to many existing Federated Learning methods, making it easily integrated with them to further enhance model performance. To validate the effectiveness of our proposed FedLWS, we conduct extensive experiments under diverse scenarios. We observe that FedLWS can significantly improve the performance of existing FL algorithms.

The contributions of this work are summarized as follows:

- We empirically show that the ratio of regularization to optimization terms in FL model aggregation is positively correlated with local gradient variance. This allows us to directly calculate the shrinking factor, eliminating the need for fine-tuning on a proxy dataset.

- We experimentally demonstrate that the model-wise shrinking factor is suboptimal to improve the generalization of the global model. It is essential to consider the discrepancy between different layers.

- We propose FedLWS, a simple method that generates the global model with better generalization through layer-wise weight shrinking. FedLWS requires neither additional data nor transmission of the original data, thus raising no privacy concern.

- We conduct extensive experiments under diverse scenarios to demonstrate that FedLWS brings considerable accuracy gains over the state-of-the-art FL approaches.

## 2 RELATED WORKS

Federated learning (FL) is a rapidly advancing research field with many remaining open problems to address. FedAvg (McMahan et al., 2017) has been the standard algorithm of FL. It trains local models separately and conducts model aggregation based on the data size. However, the distribution heterogeneity of local datasets may significantly degrade FL's performance. In this paper, we aim to attain a more effective and better generalized global model through collaborative training on both the client and server sides. Many previous works focus on this can be mainly divided into two directions.

### 2.1 CLIENT-SIDE DRIFT ADJUSTMENT

Due to the data heterogeneity, local models trained on the clients may exhibit different degrees of bias, thereby affecting the performance of the global model. Many methods aim to reduce this bias by adjusting the training process of local models. FedProx (Li et al., 2020b) utilizes the $l_2$-distance between the global model and the local model as a regularization term during the training of the local model. FedDyn (Acar et al., 2021) proposes a dynamic regularizer for each client to align the global and local solutions. MOON (Li et al., 2021) aligns the features of global and local models through contrastive learning. FedDC (Gao et al., 2022) and SCAFFOLD (Karimireddy et al., 2020) adjust the drift in the local model by introducing control variates. FedETF (Li et al., 2023b) employs a fixed

ETF classifier during training to learn unified feature representations. These methods concentrate on adjusting local models, merely assigning aggregation weights based on the size of the local dataset. In contrast, our approach focuses on the server-side aggregation process, which can be easily combined with these methods to enhance model performance further due to its flexibility.

## 2.2 SERVER-SIDE AGGREGATION SCHEME

Several other previous works adjust the server-side model by improving the model aggregation stage or incorporating a fine-tuning step to obtain a better global model. FedAvgM (Hsu et al., 2019) adopts momentum updates on the server-side to stabilize the training process. CCVR (Luo et al., 2021) adjusts the global model's classifier using virtual representations. FedDF (Lin et al., 2020) and FedBE (Chen & Chao, 2021) fine-tune the global model on the server. FedDisco (Ye et al., 2023) leverages both dataset size and the discrepancy between local and global category distributions to determine more distinguishing aggregation weights. Several prior studies have investigated layer-wise model aggregation. FedLAMA (Lee et al., 2023) adjusts the aggregation frequency for each layer to reduce communication costs while accounting for inter-layer differences. pFedLA (Ma et al., 2022) focuses on personalized FL, it designs a hyper-network to predict the layer-wise aggregation weights for each client. L-DAWA (Rehman et al., 2023) employs cosine similarity between local models and the global model as aggregation weights for model aggregation, simultaneously taking into account variations across different layers of the model. In comparison to these methods, our method is neither focused on aggregation frequency adjustment nor layer-wise aggregation weight adjustment. Instead, we propose an adaptive layer-wise weight shrinking step after model aggregation to mitigate aggregation bias, which is both computationally efficient and modular, enabling seamless integration with various FL frameworks and baselines. Recently, FedLAW (Li et al., 2023a) identifies the global weight shrinking phenomenon and then learns the optimal shrinking factor $\gamma$ and the aggregation weights $\lambda$ at the server with a proxy dataset, which is assumed to have the same distribution as the test dataset. Its success is inseparable from the high degree of consistency between the proxy data and the test data. Considering data privacy is a significant concern in FL, obtaining the proxy dataset with a distribution identical to the test dataset in practice is challenging, limiting its application in real-world. In addition, FedLAW ignores the variations across different layers of model for model aggregation. In contrast to FedLAW, our FedLWS calculate the shrinking factors directly through the easily available gradient and parameters of the global model, which takes into account the layer-wise differences, avoiding demanding proxy dataset and optimization. Moving beyond FedLAW, ours can be easily integrated with most of related model aggregation methods for decoupling shrinking factor and aggregation weights.

## 3 BACKGROUND

Federated Learning consists of $K$ clients and a central server, where each client has its own private local dataset $\mathcal{D}_k$. FL aims to enable clients to collaboratively learn a global model for the server without data sharing. In communication round $t$, the parameters of the global model and the client $k$'s model are denoted as $\mathbf{w}_g^t$ and $\mathbf{w}_k^t$, respectively. The workflow of the basic FL method, FedAvg (McMahan et al., 2017), in communication round $t$ can be described as follows:

- **Step 1:** Server broadcasts the parameters of global model $\mathbf{w}_g^t$ to each client;
- **Step 2:** Each client $k$ performs $E$ epochs of local model training on private dataset $\mathcal{D}_k$ to obtain a local model $\mathbf{w}_k^t$;
- **Step 3:** Clients upload the local models to the server;
- **Step 4:** Server merges the local models to get a new global model: $\mathbf{w}_g^{t+1} = \sum_{k=1}^{K} \lambda_k \mathbf{w}_k^t$, where $\lambda_k$ is the aggregation weight of the client $k$ and FedAvg sets $\lambda_k = \frac{|\mathcal{D}_k|}{\sum_{i=1}^{K} |\mathcal{D}_i|}$.

Denote $\mathbf{g}_k^t$ as the local gradient for the model of the $k$-th client during communication round $t$, where the local model $\mathbf{w}_k^t$ equals to $\mathbf{w}_g^t - \mathbf{g}_k^t$ in Step 2. Therefore, the update process for the global model can be expressed as follows:

$$\mathbf{w}_g^{t+1} = \mathbf{w}_g^t - \eta_g \sum_{k=1}^{K} \lambda_k \mathbf{g}_k^t, \text{ s.t. } \lambda_k \geq 0, \|\boldsymbol{\lambda}\|_1 = 1, \tag{1}$$

where $\boldsymbol{\lambda} = [\lambda_1, ..., \lambda_K]$ is the aggregation weights and $\eta_g$ is the global learning rate. FedLAW (Li et al., 2023a) identifies the global weight shrinking phenomenon during the model aggregation in FL, and introduces a global shrinking factor $0 < \gamma < 1$, which improves generalization. The model aggregation process is then reformulated as follows:

$$\mathbf{w}_g^{t+1} = \gamma^t \sum_{k=1}^{K} \lambda_k \mathbf{w}_k^t, \text{ s.t. } 1 > \gamma^t > 0, \lambda_k \geq 0, \|\boldsymbol{\lambda}\|_1 = 1, \tag{2}$$

where the $\gamma^t$ is the shrinking factor in the communication round $t$. Therefore, the right of Eq. 1 can be rewritten as:

$$\mathbf{w}_g^{t+1} = \gamma^t(\mathbf{w}_g^t - \eta_g \mathbf{g}_g^t) = \mathbf{w}_g^t - \gamma^t \eta_g \mathbf{g}_g^t - (1 - \gamma^t)\mathbf{w}_g^t, \tag{3}$$

where $\mathbf{g}_g^t = \sum_{k=1}^{K} \lambda_k \mathbf{g}_k^t$ is the global gradient. We refer to the $\gamma^t \eta_g \mathbf{g}_g^t$ as the global averaged gradient (optimization term) and $(1 - \gamma^t)\mathbf{w}_g^t$ is the pseudo gradient of global weight shrinking (regularization term). Therefore, the value of $\gamma^t$ determines the strength of regularization, with smaller values of $\gamma^t$ corresponding to stronger regularization (weight shrinking) and smaller gradient updates. It can be seen that the global weight shriking is analogous to weight decay, however they are are distinct. More discussions about their differences can be found in Section 5.3 and Appendix A.2.

FedLAW (Li et al., 2023a) learns the optimal shrinking factor $\gamma$ and the aggregation weights $\boldsymbol{\lambda}$ on a proxy dataset, which is assumed to have the same distribution as the test global dataset. Nevertheless, the utilization of a proxy dataset may raise privacy-related concerns, and obtaining a dataset with a distribution identical to the test data in real-world applications can be challenging. Moreover, numerous previous studies (Ma et al., 2022; Luo et al., 2021; Rehman et al., 2023) have illustrated that the different layers of deep neural networks exhibit different levels of heterogeneity during the training process of FL, especially when training with non-IID data. However, FedLAW(Li et al., 2023a) does not consider the discrepancy among different layers, assigning a single shrinking factor $\gamma$ to all layers of the model, which could impact the global model performance negatively, as an inappropriate value of $\gamma$ can affect the generalization. Therefore, this paper aims to address the challenge of learning an appropriate $\gamma$ without relying on a proxy dataset, while also adaptively determining the corresponding $\gamma$ value for each layer.

## 4 METHOD

In this section, we propose Federated Learning with Adaptive Layer-wise Weight Shrinking (FedLWS), which dynamically computes the layer-wise shrinking factor for each layer of the global model. Unlike prior approaches, FedLWS does not rely on a proxy dataset, thus addressing the privacy and practical deployment issues present in traditional Federated Learning (FL) systems. Additionally, our method seamlessly integrates into existing FL methods to enhance performance.

### 4.1 FEDERATED LEARNING WITH ADAPTIVE WEIGHT SHRINKING

We first analyze the balance between regularization and optimization. From Equation 3, we can see that the shrinking factor $\gamma^t$ governs the trade-off between regularization (the model tends to keep its current weights) and optimization (the model tends to update its weights according to the gradients). A smaller shrinking factor $\gamma^t$ imposes stronger regularization, while a larger $\gamma^t$ favors optimization by allowing larger updates. The challenge lies in finding an appropriate $\gamma^t$ that dynamically adapts to the training process. An ideal $\gamma^t$ should be able to maintain a balance between the regularization term and the optimization term. To address this problem, we introduce an intuitive hypothesis: the balance between the regularization term and the optimization term should be related to the variance of the gradients of local client models. Specifically, if local client models agree less with each other in terms of the gradient updates (i.e., higher variance of their gradients), the global model should be more conservative in terms of updating its weights, meaning that a stronger regularization is needed (i.e., a smaller $\gamma_t$). Conversely, if the gradient variance is smaller, one should update the global model's weights more aggressively according to the gradients (less regularization and higher $\gamma_t$). Specifically, denoting the gradient variance of the local models as $\tau^t$, we formulate it as follows:

$$\tau^t = \frac{1}{K} \sum_{k=1}^{K} \|\mathbf{g}_k^t - \mathbf{g}_{mean}^t\|, \ \mathbf{g}_{mean}^t = \frac{1}{K} \sum_{k=1}^{K} \mathbf{g}_k^t \tag{4}$$

where $\mathbf{g}_k^t$ is the local gradient of the $k$-th client. Therefore, given the discussion above, we assume the gradient variance of the local models is proportional to the balance between the regularization and optimization term, expressed as:

$$\tau^t \propto r^t, r^t = \frac{(1-\gamma^t)\|\mathbf{w}_g^t\|}{\gamma^t\|\eta_g\mathbf{g}_g^t\|}, \text{ s.t. } 0 < \gamma^t < 1, \tag{5}$$

To empirically substantiate this hypothesis, we conducted the experiments employing the test dataset to optimize $\gamma^t$, which is impractical in real-world cases but for demonstrating purpose here. After the optimal value of $\gamma^t$ is obtained, we compute the right-hand-side term of Equation 5 and denote its value as $r^t$. For the left-hand-side term of Equation 5, $\tau^t$, we compute its value with Equation 4. To test the correlation between $r^t$ and $\tau^t$ with different magnitudes of data heterogeneity, we adopt Dirichlet sampling $Dir_\alpha$ to simulate client heterogeneity. A smaller value of $\alpha$ indicates a higher degree of non-IID data distribution. The experimental results are illustrated in Figure 1(a). It can be observed that as the degree of data heterogeneity diminishes, both $r$ and $\tau$ exhibit a corresponding reduction. This suggests that, in the case of IID data, $\gamma$ tends to facilitate larger updates. More exploration and theory analysis can be found in Section 5.3, Appendix B.2 and D. We also compute Pearson correlation coefficient (Cohen et al., 2009) between $r$ and $\tau$, which is 0.860. Together with the analysis of Figure 1(a), it validates our hypothesis.

We further introduce the scaling term $\beta$ as a hyperparameter:

$$\beta\tau^t = \frac{(1-\gamma^t)\|\mathbf{w}_g^t\|}{\gamma^t\|\eta_g\mathbf{g}_g^t\|}, \text{ s.t. } 0 < \gamma^t < 1. \tag{6}$$

Combining Equation 6 with Equation 4, we can readily deduce the computational expression for the shrinking factor $\gamma^t$ as follows:

$$\gamma^t = \frac{\|\mathbf{w}_g^t\|}{\beta\tau^t\|\eta_g^t\mathbf{g}_g^t\| + \|\mathbf{w}_g^t\|}. \tag{7}$$

During the model aggregation process, both $\mathbf{w}_g^t$ and $\eta_g\mathbf{g}_g^t = \widehat{\mathbf{w}}_g^{t+1} - \mathbf{w}_g^t$ are known. Therefore, after calculating $\tau^t$ via Equation 4, the value of shrinking factor $\gamma^t$ can be directly obtained through Equation 7. Equation 7 enables a comprehensive consideration of both the global gradient and model parameters, ensuring a balanced interplay between the optimization and regularization terms during the FL training process and eliminating the need for optimization using additional proxy datasets.

FedLWS is conducted after model aggregation, making it easily be combined with other Federated Learning methods. For example, when combined with FedAvg, the first four steps of our method align with the **Step 1** $\sim$ **Step 4** in Section 3. Following this, our method computes the shrinking factors and applies layer-wise weight shrinking to the aggregated model. In communication round $t$, we use $\widehat{\mathbf{w}}_g^{t+1} = \sum_{k=1}^K \lambda_k \mathbf{w}_k^t$ represent the aggregated model before weight shrinking and denote the aggregated model after weight shrinking as $\mathbf{w}_g^{t+1}$.

## 4.2 LAYER-WISE EXTENSION OF FEDLWS

Now, we investigate the effect of layer-wise factors on the performance of the global model. Previous studies have demonstrated divergence across various layers of deep neural networks (Ma et al., 2022; Rehman et al., 2023; Lee et al., 2023). Therefore, applying a single shrinking factor, $\gamma$, to the entire model may not fully capture the benefits of global weight reduction. We validate this through experiments, as shown in Figure 1(b), the model we used is a 5-layer CNN, the layer-wise $\gamma$ refers to employing distinct shrinking factors for each layer of the model, with values uniformly decreasing from 1 to 0.96. Figure 1(b) illustrates that considering $\gamma$ in a layer-wise manner, rather than using a single $\gamma$ for the entire model, can further enhance the model's generalization performance.

Considering that each layer in deep neural networks may vary differently, it might be beneficial to design a respective regularization strength for each layer of the global model. Therefore, we aim to calculate the shrinking factor for each layer, using $\gamma_l^t$ to represent the shrinking factor of the $l$-th layer of the global model in communication round $t$. Benefiting from the Equation 4 and 7, we can achieve the goal by calculating $\gamma_l^t$ as follows:

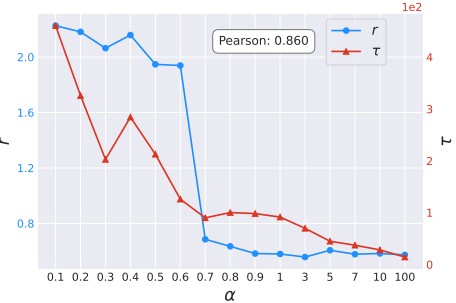 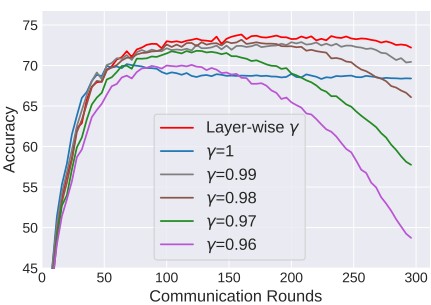

(a) Gradient variance $\tau$ and ratio $r$ under different degrees of data heterogeneity.

(b) Accuracy curves with different fixed $\gamma$.

Figure 1: Empirical observations on CIFAR-10 with CNN as the backbone; see more results in Appendix B.2. In (a), $\alpha$ is the degree of data heterogeneity, with smaller $\alpha$ indicating more heterogeneous data, it illustrates the correlation between $\tau$ and ratio. (b) indicates that layer-wise $\gamma$ can enhance the global model's performance.

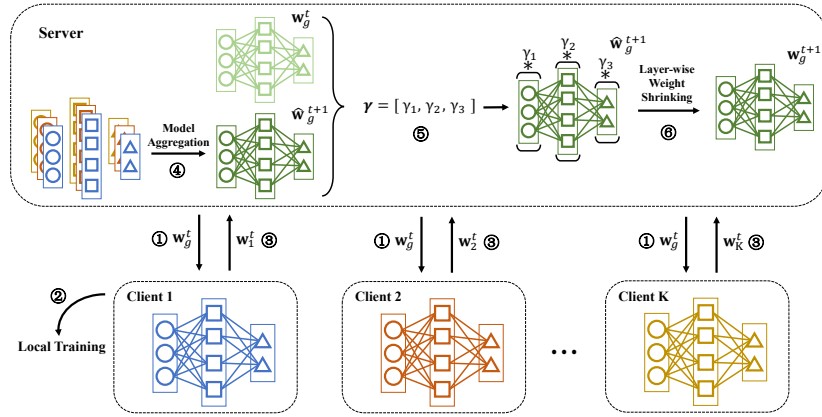

Figure 2: The overview of FedLWS. ① server broadcasts the parameters of the global model to each client; ② clients perform local training on their own dataset; ③ clients upload the local models to the server; ④ the server conduct model aggregation to generate new global model $\widehat{\mathbf{w}}_g^{t+1}$; ⑤ the server calculates the layer shrinking factors $\boldsymbol{\gamma}$; ⑥: the serve conduct layer-wise weight shrinking and obtain the final global model $\mathbf{w}_g^{t+1}$.

$$\gamma_l^t = \frac{\|\mathbf{w}_{gl}^t\|}{\beta \tau_l^t \|\eta_g^t \mathbf{g}_{gl}^t\| + \|\mathbf{w}_{gl}^t\|}, \quad \tau_l^t = \frac{1}{K} \sum_{k=1}^{K} \|\mathbf{g}_{kl}^t - \mathbf{g}_{meanl}^t\|, \quad (8)$$

where $\mathbf{g}_{gl}^t$ and $\mathbf{w}_{gl}^t$ represent the gradients and parameters of the $l$-th layer of the global model. Through Equation 8, the server obtains the layer-wise shrinking factors $[\gamma_1^t, \gamma_2^t, ..., \gamma_L^t]$, and then conducts layer-wise weight shrinking to generate the new global model $\mathbf{w}_g^{t+1}$ as follows:

$$\mathbf{w}_g^{t+1} = \gamma_L^t(\widehat{\mathbf{w}}_{gL}^{t+1}) \circ ... \circ \gamma_2^t(\widehat{\mathbf{w}}_{g2}^{t+1}) \circ \gamma_1^t(\widehat{\mathbf{w}}_{g1}^{t+1}), \quad (9)$$

where the "$\circ$" denotes the connection between different layers within the model, $\widehat{\mathbf{w}}_{gl}^{t+1}$ is the $l$-th layer's parameter of $\widehat{\mathbf{w}}_g^{t+1}$. Having introduced our adaptive way of setting $\gamma_l^t$, we give an overview of our method, FedLWS, in Figure 2. The pseudo-code of FedLWS is shown in Appendix C.1, Algorithm 1, where we highlight the additional steps required by our method compared to FedAvg.

## 4.3 DISCUSSIONS

**Privacy.** Our proposed FedLWS is more privacy-preserving and adaptable in practice compared to some of the previous works (Luo et al., 2021; Lin et al., 2020; Li et al., 2023a). FedLWS directly calculates the value of the shrinking factors $\gamma$ without the need for a proxy dataset or a fine-tuning step. The calculation of the shrinking factor $\gamma$ is based on the parameters and the gradient updates of the global model, which can be easily obtained by the server during the training process, without the need to transmit any additional information. Hence, our proposed FedLWS avoids data leakage and is more applicable to real-world scenarios.

Table 1: Top-1 test accuracy (%) on four datasets with three different degrees of heterogeneity.

| Dataset | FashionMNIST | | | CIFAR-10 | | | CIFAR-100 | | | Tiny-ImageNet | | | Average |
|---|---|---|---|---|---|---|---|---|---|---|---|---|---|
| Heterogeneity | $\alpha$=100 | $\alpha$=0.5 | $\alpha$=0.1 | $\alpha$=100 | $\alpha$=0.5 | $\alpha$=0.1 | $\alpha$=100 | $\alpha$=0.5 | $\alpha$=0.1 | $\alpha$=100 | $\alpha$=0.5 | $\alpha$=0.1 | |
| FedLAW (Li et al., 2023a) | 89.78 | 89.23 | 87.52 | 81.30 | 75.27 | 64.76 | 41.05 | 37.56 | 34.59 | 37.20 | 33.49 | 29.13 | 58.41 |
| FedAvg (McMahan et al., 2017) | 90.44 | 90.04 | 88.62 | 76.01 | 74.47 | 61.04 | 41.46 | 37.21 | 36.71 | 36.31 | 34.43 | 29.44 | 58.02 |
| +LWS (Ours) | **90.99** | **90.33** | **88.99** | **76.85** | **75.63** | **64.08** | **42.42** | **41.03** | **37.70** | **37.16** | **35.12** | **31.34** | **59.30** |
| FedDisco (Ye et al., 2023) | 90.68 | 89.90 | 88.54 | 75.40 | 74.72 | 62.86 | 41.46 | 37.28 | 36.46 | 35.92 | 34.29 | 29.39 | 58.16 |
| +LWS (Ours) | **90.95** | **90.43** | **89.39** | **76.34** | **75.41** | **66.91** | **42.68** | **41.68** | **36.93** | **37.30** | **34.92** | **31.53** | **59.54** |
| L-DAWA (Rehman et al., 2023) | 89.97 | 86.01 | 79.03 | 75.37 | 75.61 | 62.87 | 42.38 | 39.81 | 36.31 | 33.55 | 31.43 | 30.02 | 49.30 |
| +LWS (Ours) | **91.05** | **86.63** | **86.11** | **76.21** | **76.77** | **65.13** | **42.97** | **40.40** | **37.04** | **36.83** | **33.77** | **32.24** | **51.19** |
| FedProx (Li et al., 2020b) | 91.24 | 90.69 | 88.78 | 73.96 | 73.27 | 60.62 | 38.15 | 39.35 | 34.60 | 35.03 | 34.32 | 29.37 | 57.45 |
| +LWS (Ours) | **91.35** | **91.24** | **89.25** | **74.34** | **74.55** | **62.54** | **38.64** | **39.93** | **35.37** | **35.29** | **34.98** | **30.68** | **58.18** |
| FedDyn (Acar et al., 2021) | 90.29 | 88.57 | 87.84 | 77.92 | 74.61 | 56.37 | 41.04 | 44.80 | 36.92 | 34.32 | 32.80 | 27.32 | 57.73 |
| +LWS (Ours) | **90.69** | **89.48** | **88.26** | **78.90** | **77.33** | **61.61** | **46.14** | **46.38** | **37.22** | **34.88** | **33.20** | **27.74** | **59.32** |

**Modularity.** Our proposed FedLWS can be a plug-and-play module in many existing FL methods to further improve their performance. It has a broad range of applications. For the FL methods adjusting the client-side model (Li et al., 2020b; Acar et al., 2021; Li et al., 2021), FedLWS operates on the server-side, therefore it can be easily integrated with these client-side adjustment methods. Moreover, for the methods adjusting the server-side model, our FedLWS is conducted after the model aggregation, and most previous works (McMahan et al., 2017; Ye et al., 2023; Hsu et al., 2019) commonly normalize the aggregation weights (the sum of weight is 1), which is orthogonal to our method. Hence, our FedLWS can also be incorporated with them by conducting weight shrinking after model aggregation. While both ours and FedLAW (Li et al., 2023a) consider the shrinking factor, FedLAW lacks flexibility in combination with other methods, as it learns not only the shrinking factor but also the aggregation weight, which is non-orthogonal with the previous work on model aggregation adjustment. Moreover, it requires a proxy dataset, a necessity not present in ours. More discussions can be found in Appendix A.

## 5 EXPERIMENTS

### 5.1 EXPERIMENT SETUP

**Dataset and Baselines.** In this paper, we consider four image classification datasets: CIFAR-10 (Krizhevsky et al., 2009), CIFAR-100 (Krizhevsky et al., 2009), FashionMNIST (Xiao et al., 2017), and Tiny-ImageNet (Chrabaszcz et al., 2017); and text classification datasets: AG News (Zhang et al., 2015), Sogou News (Zhang et al., 2015), and Amazon Review (Ben-David et al., 2006). We compare our method with six representative baselines. Among these, 1) FedAvg (McMahan et al., 2017) is the standard algorithm of Federated Learning; 2) FedProx (Li et al., 2020b) and FedDyn (Acar et al., 2021) focus on the adjustment of the local model; 3) Feddisco (Ye et al., 2023) and L-DAWA (Rehman et al., 2023) focus on the adjustment of the aggregation scheme. Our approach can be easily integrated with the methods mentioned above. Additionally, we also demonstrate the performance of FedLAW (Li et al., 2023a). However, since it leverages additional data for fine-tuning that other methods do not, we present it only for reference. In the experiments, our method FedLWS is the layer-wise approach described in Section 4.2 unless indicated specifically.[1]

Table 2: Results on text classification datasets.

| Method | With LWS? | AG News | | Sogou News | | Amazon Review |
|---|---|---|---|---|---|---|
| | | $\alpha = 0.1$ | $\alpha = 0.5$ | $\alpha = 0.1$ | $\alpha = 0.5$ | Feature Shift |
| FedAvg | × | 73.43 | 70.37 | 87.68 | 91.53 | 88.15 |
| | √ | **74.96** | **72.32** | **90.56** | **92.76** | **88.62** |
| FedProx | × | 65.07 | 74.56 | 88.60 | 92.28 | 88.24 |
| | √ | **75.24** | **77.18** | **90.17** | **93.10** | **88.75** |

Table 3: Average aggregation execution time (Sec) across different model structures.

| Method | CNN | ResNet20 | ViT | WRN56_4 | DenseNet121 |
|---|---|---|---|---|---|
| FedAvg | 0.019 | 0.10 | 0.18 | 0.561 | 1.359 |
| FedLAW | 4.830 | 7.11 | 9.80 | 20.08 | 27.25 |
| **FedLWS (Ours)** | **0.035** | **0.12** | **0.21** | **0.832** | **1.756** |

**Federated Simulation.** To emulate the FL scenario, we randomly partition the training dataset into $K$ groups and assign group $k$ to client $k$. Namely, each client has its local training dataset. We reserve the testing set on the server-side for evaluating the performance of global model. In practical FL scenarios, the clients often exhibit heterogeneity, leading to non-IID characteristics among their data. In this paper, we employ Dirichlet sampling $Dir_\alpha$ to synthesize client heterogeneity, it is widely used in FL literature (Wang et al., 2020a; Yurochkin et al., 2019; Ye et al., 2023). The smaller the value of $\alpha$, the greater the non-IID. We apply the same data synthesis approach to all methods for a fair comparison. More implementation details can be found in Appendix C.

---

[1]The source code is available at `https://github.com/ChanglongShi/FedLWS`

## 5.2 PERFORMANCE EVALUATION

**Performance and Modularity.** In this section, we consider six representative methods and report the test accuracy on all datasets before and after applying our FedLWS under different heterogeneity settings in Table 1. It can be observed that the application of FedLWS leads to increased accuracies for all methods across various datasets and heterogeneity settings, highlighting the effectiveness of our proposed method. This is particularly inspiring because FedLWS necessitates no modification to the original federated training process. The accuracy enhancements can be easily attained by simply post-processing the aggregated global model. As shown in Table 1, our method performs better on more complex or heterogeneous data. This can be explained as follows: FedAvg already achieves strong results on relatively simple tasks or datasets with IID distributions (Li et al., 2020b; Ye et al., 2023). In such scenarios, the differences between client models are relatively small, and consequently, the $\gamma$ computed using Equation 7 tends to be closer to 1. This means that FedLWS's behavior aligns more closely with the baseline under these conditions. However, on datasets with greater complexity or heterogeneity, client model differences become more pronounced. FedLWS effectively addresses these differences through layer-wise weight shrinking, resulting in improved global model performance. Furthermore, it can be observed that FedLAW performs better on the CIFAR-10 dataset and, in some scenarios, even surpasses our method. One possible explanation is the proxy dataset used in our experiments, which contains 200 samples, consistent with the original settings in (Li et al., 2023a). For CIFAR-10, this provides 20 samples per class, providing sufficient information to optimize the shrinking factor effectively. In contrast, for datasets like CIFAR-100 and TinyImageNet, the proxy dataset contains only 2 and 1 sample per class, respectively. This limited representation makes it difficult for FedLAW to train an optimal shrinking factor, which may explain the variations in its performance across different scenarios. To verify that our method can also be applied to text modality, we conducted experiments on NLP datasets under different heterogeneity settings. Table 2 shows that FedLWS still consistently improves the baselines on text modality.

**Computation Efficiency.** In Table 3, we show the aggregation execution time of FedAvg, FedLWS, and the closely related work FedLAW (Li et al., 2023a) across various model architectures. For FedLAW, the proxy dataset contains 200 samples and the server epoch is 100 (consistent with (Li et al., 2023a)). It can be observed that, in comparison to FedLAW, our FedLWS requires significantly less execution time, as FedLAW necessitates an additional proxy dataset for fine-tuning, which we do not require. Although larger models may slightly increase the computational load, our method remains markedly more efficient than optimization-based alternatives, further demonstrating the efficiency of our approach.

Table 4: Comparison of layer-wise and model-wise weight shrinking effects.

| Dataset | CIFAR-10 | | CIFAR-100 | |
|---|---|---|---|---|
| **Heterogeneity** | $\alpha$=100 | $\alpha$=0.1 | $\alpha$=100 | $\alpha$=0.1 |
| FedAvg | 76.01 | 61.04 | 41.46 | 36.71 |
| FedLWS(Model-wise) | 76.17 | 63.21 | 41.77 | 37.65 |
| FedLWS(Layer-wise) | **76.85** | **64.08** | **42.42** | **37.70** |

Table 5: The performance of compared methods with different model architectures.

| Model | ResNet20 | WRN56_4 | DenseNet121 | ViT | Dataset |
|---|---|---|---|---|---|
| FedLAW | 75.72 | 80.46 | 86.43 | 51.20 | |
| FedAvg | 75.07 | 78.97 | 86.14 | 51.31 | CIFAR-10 |
| +LWS | **76.17** | **81.29** | **86.63** | **54.14** | |
| FedLAW | 36.53 | 36.60 | 55.36 | 22.03 | |
| FedAvg | 36.71 | 39.71 | 56.59 | 25.60 | CIFAR-100 |
| +LWS | **38.01** | **42.37** | **57.41** | **26.05** | |

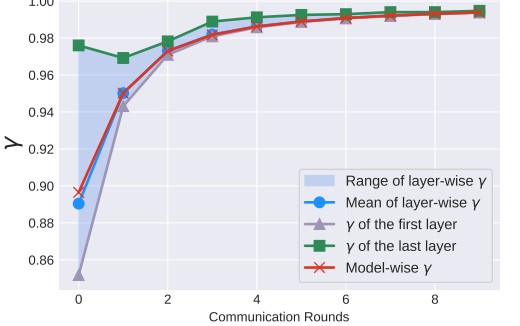

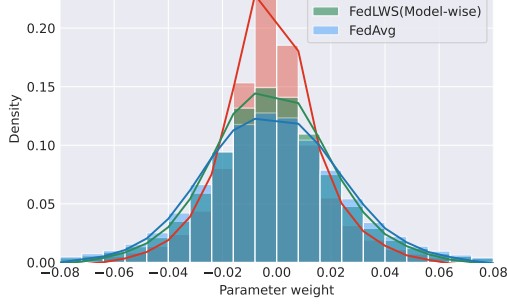

(a) The value of model-wise and layer-wise shrinking factors during the training process ($\alpha$=0.1).

(b) The histogram of final models' parameters.

Figure 3: The comparison between layer-wise FedLWS and model-wise FedLWS (ResNet20).

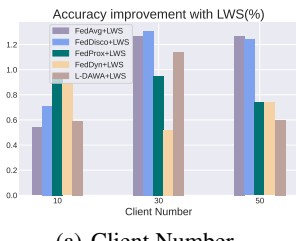 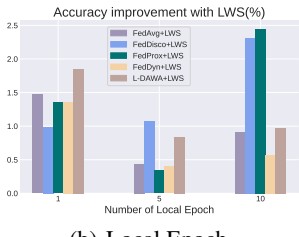 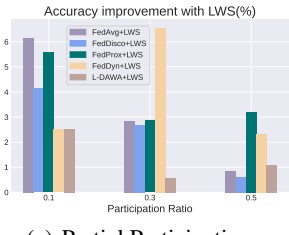

(a) Client Number.                    (b) Local Epoch.                    (c) Partial Participation.

Figure 4: Accuracy improvement after applying LWS under different client numbers, local epochs, and partial participation ratios.

## 5.3 ABLATION STUDIES

**Effects of layer-wise shrinking factor.** In this section, we conduct experiments to analyze the effectiveness of the layer-wise shrinking factor. As a point of comparison, we implement the model-wise FedLWS by computing a model-wise shrinking factor $\gamma$ for the entire model, which can be viewed as a degraded version of layer-wise FedLWS. The model-wise shrinking factor is calculated according to Equation 7, based on the entire model's parameters and gradient. In Table 4, taking the FedAvg as the baseline, we compare the performance of layer-wise and model-wise weight shrinking across different datasets and heterogeneity degrees. Table 4 demonstrates that layer-wise weight shrinking achieves higher model accuracy compared to model-wise weight shrinking, indicating that the layer-wise approach allows for more precise adjustments in the model aggregation process, thereby yielding improved results. In addition, even introducing the degraded model-wise shrinking factor into FedAvg can still achieve performance improvement, suggesting the effect of shrinking factor on the global model's generalization ability.

In Figure 3(a), we compare the values of model-wise $\gamma$ and layer-wise $\gamma$ during the training process in Resnet20, where we show the mean of layer-wise $\gamma$ of all layers, $\gamma$ at the first layer and $\gamma$ at the last layer due to the limited space. The variation of layer-wise $\gamma$ for each layer is deferred to Figure 9 in Appendix. It can be observed that throughout the training process, the value of model-wise $\gamma$ and the mean of layer-wise $\gamma$ are very close. Besides, the variance of the layer-wise $\gamma$ gradually decreases with the training process. That is to say, at the initial iterations, there is a significant difference in layer-wise $\gamma$ among different layers of the global model. As the training progresses, the gap between layers continues to diminish. This indicates that the difference between layer-wise $\gamma$ and model-wise $\gamma$ primarily occurs at the initial stages of training. Layer-wise $\gamma$ can better adjust the global model at the initial training period. It can be inferred that layer-wise $\gamma$ can enhance better utilization of the global weight shrinking phenomenon and improve model generalization by assigning optimal $\gamma$ values for each layer of the global model. We show the histogram of the final models' parameters in Figure 3(b). It can be seen that, in comparison to FedAvg and model-wise FedLWS, layer-wise FedLWS makes more model parameters close to zero, which is similar to weight decay. This may explain why FedLWS enhances the generalization ability of the global model. More results about shrinking factor can be found in Appendix B.

**Effects of model architectures.** In Table 5, we evaluate our proposed FedLWS across various model architectures, including ResNet (He et al., 2016), Wide-ResNet (WRN) (Zagoruyko & Komodakis, 2016), DenseNet (Huang et al., 2017) and Vision Transformer (ViT) (Dosovitskiy et al., 2020). The results demonstrate the effectiveness of FedLWS across different model architectures, indicating its robust performance even as the network depth or width increases.

**Effects of client number, local epoch, and partial participation.** In this section, we tune three crucial parameters in FL: the number of clients $K \in \{10, 30, 50\}$, the number of local epoch $E \in \{1, 5, 10\}$, and partial participation ratio $R \in \{0.1, 0.3, 0.5\}$. We show the accuracy improvement brought by FedLWS in Figure 4(a), 4(b), and 4(c), respectively. The experiments consistently reveal that our proposed method consistently brings performance improvement across different FL settings.

**Relation with Weight Decay.** While our method differs from weight decay, it is still meaningful to compare it with adaptive weight decay methods. Therefore, we applied two adaptive weight decay methods, AWD (Ghiasi et al., 2023) and AdaDecay (Nakamura & Hong, 2019), to Federated Learning model aggregation process, and compared them with our approach, the result as shown in

Table 6: Top-1 test accuracy (%) on CIFAR-10 and CIFAR-100 with different degrees of heterogeneity. Compared with weight decay methods.

| Dataset | CIFAR-10 | | | CIFAR-100 | | | Average |
|---|---|---|---|---|---|---|---|
| Heterogeneity | IID($\alpha$=100) | NIID($\alpha$=1) | NIID($\alpha$=0.1) | IID($\alpha$=100) | NIID($\alpha$=1) | NIID($\alpha$=0.1) | |
| FedAvg | 76.01 | 75.18 | 61.04 | 41.46 | 41.62 | 36.71 | 55.34 |
| +AdaDecay (Nakamura & Hong, 2019) | 76.21 (↑0.20) | 75.33 (↑0.15) | 61.34 (↑0.30) | 42.15 (↑0.69) | 41.73 (↑0.11) | 36.98 (↑0.27) | 55.62 (↑0.28) |
| +AWD (Ghiasi et al., 2023) | 76.30 (↑0.29) | 75.64 (↑0.46) | 61.15 (↑0.11) | **42.81 (↑1.35)** | 41.76 (↑0.14) | 37.17 (↑0.46) | 55.81 (↑0.47) |
| **+LWS (Ours)** | **76.85 (↑0.84)** | **75.88 (↑0.70)** | **64.08 (↑3.04)** | 42.42 (↑0.96) | **42.93 (↑1.31)** | **37.70 (↑0.99)** | **56.64 (↑1.30)** |

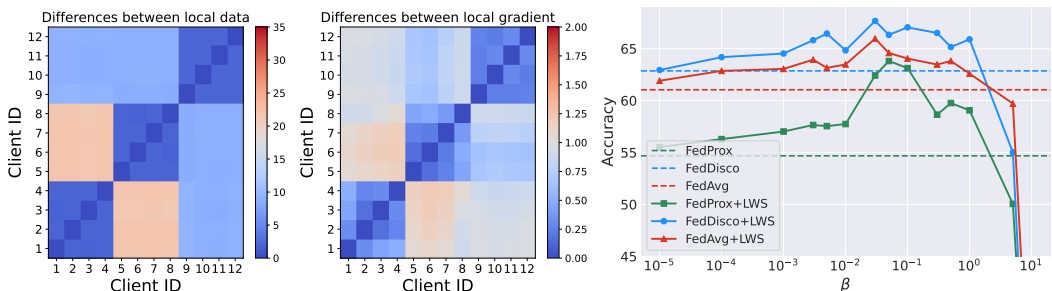

Figure 5: Divergence in local data and gradients      Figure 6: Accuracy with different $\beta$.

Table 6. It demonstrates that our method significantly outperforms the adaptive weight decay methods in the Federated Learning scenario. More discussions can be found in Appendix A.2.

**Relationship between local dataset and local gradient.** Our method uses local gradients to calculate the shrinking factor. To analyze the reasons for its effectiveness, we conducted additional experiments for further observation. In Figure 5, we illustrate the divergence between different local datasets and between the correspondinglocal gradients (i.e., local model's parameter changes after local training). As shown, the distances in local gradients closely resemble those of the local data distributions. This demonstrates that the local gradient can effectively capture relevant information about the local data. In other words, our calculation of the local gradient variance, $\tau$, also serves as a measure of the degree of data heterogeneity. For more heterogeneous data, stronger regularization is applied, making the training process more stable. More implementation details can be found in Appendix B.3.

**Hyperparameter.** In this section, we explore the impact of varying the hyperparameter $\beta$ to highlight the flexibility and adaptability of hyperparameter tuning in our proposed FedLWS. In Figure 6, we demonstrate the impact of different $\beta$ values on accuracy when FedLWS is combined with various methods. As $\beta$ approaches 0, the $\gamma$ value calculated using Equation 7 converges to 1, resulting in the model's performance degrading to that of the baseline. Conversely, when $\beta$ is too large, the calculated $\gamma$ becomes excessively small, which can cause model instability or even failure. Based on our experiments, we recommend a safe range for $\beta$ between 0.001 and 0.1.

## 6 CONCLUSION

In this paper, through empirical explorations, we show that layer-wise weight shrinking can further improve the generalization of the global model. Subsequently, we investigate the factors influencing the shrinking factor during the training process of Federated Learning. Based on these observations, we present FedLWS, a method that directly computes layer-wise shrinking factors without requiring any additional data. FedLWS can be seamlessly integrated with the existing Federated Learning methods to further enhance the performance of the global model. Experiments demonstrate that FedLWS steadily enhances the state-of-the-art FL approaches under various settings. A potential limitation of our method is that it is only applicable to scenarios where the client model architectures are identical. The heterogeneity in client model architecture is a common issue in FL, and this limitation is prevalent in many FL methods. We plan to address the application in scenarios with heterogeneous client models in future work.

ACKNOWLEDGEMENTS

We truly thank the reviewers for their great effort in our submission. Changlong Shi, Jinmeng Li, Dandan Guo and Yi Chang are supported by the National Natural Science Foundation of China (No. U2341229, No. 62306125) and the National Key R&D Program of China under Grant (No. 2023YFF0905400).

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

# A  MORE DISCUSSION

## A.1  MODULARITY.

Our proposed FedLWS can be easily incorporated with many existing FL methods to further improve their performance. For the FL methods adjusting the client-side model (Li et al., 2020b; Acar et al., 2021; Li et al., 2021), which corresponds to the line 6 in Algorithm 1, since our proposed FedLWS operates on the server-side with lines 11 and 12 in Algorithm 1, it can be easily integrated with these methods. Moreover, for the methods that conduct the server-side model adjustment, most previous works (McMahan et al., 2017; Ye et al., 2023; Hsu et al., 2019) commonly normalize the aggregation weights (the sum of weight is 1), which is orthogonal to our method. Hence, our FedLWS can also be incorporated with them by conducting weight shrinking after model aggregation, further enhancing the model's generalization performance. For instance, regardless of whether the server utilizes FedAvg (McMahan et al., 2017) or FedDisco (Ye et al., 2023) for model aggregation (i.e., achieving the global model $\widehat{\mathbf{w}}_g^{t+1}$ like line 10 in Algorithm 1), we can further strengthen the aggregation process by implementing the lines 11 and 12 in Algorithm 1.

## A.2  RELATION WITH WEIGHT DECAY.

In Table 6, AWD and AdaDecay were not originally designed for the federated learning (FL) context. In our work, we adapted these methods to make them applicable to FL settings for a fair comparison with FedLWS. However, these methods do not account for the differences between local models, which are critical in heterogeneous FL scenarios. In contrast, FedLWS explicitly addresses these differences by considering the variance of client update gradients, combined with its layer-wise adaptability, leading to better performance in FL environments. Although the layer-wise weight shrinking (LWS) effect is analogous to weight decay, these two methods are distinct. 1) LWS employs a distinctive sparse regularization frequency, modifying model weights only in each round, resulting in stronger regularization. In LWS, $1 - \gamma$ is near 0.05, which is significantly larger than the value of the weight decay (typically $10^{-4}$). 2) LWS shrinks the global model rather than decaying the model by subtracting a decay term. 3) In this paper, LWS is conducted on the server side to adjust the aggregated model. At this stage, the model is not trainable. We utilize parameter variations as pseudo-gradients to compute the shrinking factor. In contrast, weight decay is applied to the local model training process on the client side. Consequently, the two methods are not conflicted in FL.

## A.3  RELATION WITH FEDLAW.

Learning aggregation weights of local models is an effective solution to improve Federated Learning. Recently, FedLAW (Li et al., 2023a) has gradually attracted the attention of the researchers, which identifies the global weight shrinking phenomenon and then learns the optimal shrinking factor $\gamma$ and the aggregation weights $\lambda$ on the server. Despite the effectiveness of FedLAW, it needs to optimize the $\gamma$ and $\lambda$ at the server with additional proxy dataset, which is assumed to have the same distribution as the test dataset. We have shown in Table 7 that only change the class distribution of the proxy dataset can reduce the performance of FedLAW a lot, where proxy dataset and test dataset are still from the same dataset but with different class distributions. Considering data privacy is a significant concern in Federated Learning, obtaining the proxy dataset with a distribution identical to the test dataset in practice is challenging, limiting its application in real-world. The original FedLaw paper also conducted similar experiments on CIFAR-10 with $\alpha$=100 and 0.1, but compared to theirs, our experiments are more comprehensive, covering both CIFAR-10 and CIFAR-100 datasets with $\alpha$ values of 100 and 0.1, the latter indicating a higher degree of heterogeneity. Consequently, when comparing our results in Table 7 with those in Table 6 of the FedLAW paper, only the scenario with $\alpha = 100$ is directly comparable. As observed, when $\alpha = 100$, the performance drop in the original paper is 2.26% (from 79.40 to 77.14), while in our corresponding scenario, the drop is 4.61% (from 81.30 to 76.69). The performance after applying long-tailed proxy data is similar across both studies; however, the results in our experiment outperform those in the original paper when using balanced proxy data. This could be attributed to the higher quality of our randomly sampled balanced proxy data. This further highlights that the effectiveness of the FedLAW method is highly dependent on proxy data quality. Besides, FedLAW learns the shrinking factor $\gamma$ and the aggregation weights $\lambda$ at the server jointly, making it difficult to combine with other aggregation methods in FL. Last but

not least, FedLAW ignores the variations across different layers of model for model aggregation. Therefore, designing an effective and flexible method to solve above-mentioned problems is quite necessary for model aggregation in FL.

To this end, we propose a novel model aggregation strategy, Federated Learning with Layer-wise Weight Shrinking (FedLWS). It is non-trivial since we deduce the expression of the shrinking factor and can calculate it directly through the easily available gradient and parameters of the global model, which avoids demanding proxy dataset and optimization. Considering the distinctions among layers within deep models, we further design the shrinking factor in a layer-wise manner, which is feasible and effective due to the deduced expression. Besides, ours can be easily integrated with most of related model aggregation methods for decoupling shrinking factor and aggregation weights. Therefore, ours is not just applying FedLAW to each layer of the global model. Even we only consider a single shrinking factor $\gamma$ for all layers, ours is not equal to FedLAW. We have provided our degraded model-wise version (shared shrinking factor across all layers) in Table 8. It is evident that our model-wise version still performs betters than FedLAW, proving the difference between FedLAW and ours. Furthermore, since we do not need to optimize the shrinking factor on the server, ours has less computational cost than FedLAW; As shown in Table 3 of the main text, the computational time required by FedLAW is nearly 60 times that of our method. These observations and results show that our proposed method can serve as an effective plug-and-play module in many existing FL methods, without demanding proxy dataset and additional computational cost.

To summarize, FedLAW is the state-of-the-art method in the line of the optimal shrinking factor for model aggregation in FL. We propose a new method in this line that improves over FedLAW via nontrivial efforts, which we believe has significant contributions.

Table 7: Performance in scenarios where proxy data are long-tailed and test data are balanced. The numbers in parentheses indicate the decrease compared to using proxy data with the same distribution as the test data.

| Dataset | Cifar10($\alpha = 0.1$) | Cifar10($\alpha = 100$) | Cifar100($\alpha = 0.1$) | Cifar100($\alpha = 100$) | Avg |
|---|---|---|---|---|---|
| **FedLAW (LT)** | 56.91($\downarrow$7.85) | 76.69($\downarrow$4.61) | 27.57($\downarrow$7.02) | 36.58($\downarrow$4.47) | 49.44($\downarrow$5.99) |
| **FedAvg+LWS (Ours)** | **64.08** | **76.85** | **37.70** | **42.42** | **55.26** |

Table 8: Evaluation of model-wise methods ($\alpha = 0.1$).

| Dataset | Fmnist | Cifar100 | Cifar10 | TinyImageNet | Avg |
|---|---|---|---|---|---|
| FedLAW (Li et al., 2023a) | 87.52 | 34.59 | 64.76 | 29.13 | 54.00 |
| Ours (Model-wise) | 87.95 | 36.41 | 65.31 | 30.39 | 55.02 |
| Ours (Layer-wise) | **89.39** | **36.93** | **66.91** | **31.53** | **56.19** |

## B    EXTRA EXPERIMENTAL RESULTS

### B.1    EXPERIMENTS ON PRETRAINED MODEL.

In this section, we conducted experiments using a pretrained ResNet20 on various datasets and degrees of data heterogeneity. The experimental results as shown in Table 9. It can be seen that when using a pretrained model for initialization, FedLWS can still further enhance the performance of the global model. This demonstrates the effectiveness of our method. When the global model is initialized using a pretrained model, the shrinking factor of FedLWS is close to 1 (around 0.99) during the training process.

Table 9: The performance of FedLWS using a pretrained ResNet20 model for initialization.

| Dataset | CIFAR-10 | | CIFAR-100 | |
|---|---|---|---|---|
| **Heterogeneity** | $\alpha = 0.1$ | $\alpha = 1$ | $\alpha = 0.1$ | $\alpha = 1$ |
| FedAvg | 86.74 | 91.30 | 63.98 | 66.77 |
| **+LWS (Ours)** | **87.02** | **91.47** | **64.14** | **67.06** |

This can be attributed to the superior performance of the pretrained model, which requires only minimal adjustments during the training process. As a result, there are smaller gradient updates and less necessity for strong regularization. Therefore, when the global model is initialized using a

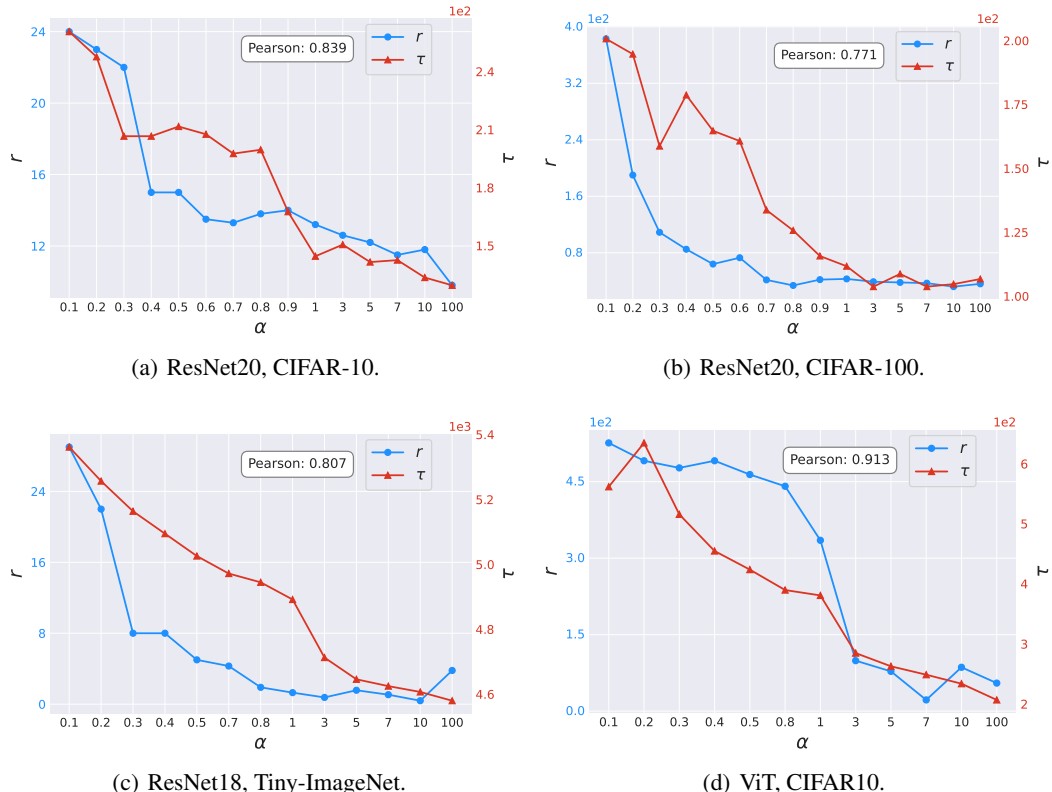

Figure 7: Gradient variance $\tau$ and ratio $r$ under different degrees of data heterogeneity.

pretrained model, our method exhibits limited influence on improving the global model. This finding is consistent with the insights presented in our paper.

### B.2 RELATION BETWEEN THE GRADIENT VARIANCE AND THE BALANCE RATIO

To further validate our hypothesis, in addition to the experiments shown in Figure 1(a), we conducted additional experiments on different datasets and model architectures. The results, as shown in Figure 7, indicate a positive correlation between gradient variance $\tau$ and the balance ratio $r$ across different model architectures and datasets, further supporting our hypothesis. It can be observed that both $r$ and $\tau$ exhibit a corresponding reduction as the degree of data heterogeneity diminishes. It is also consistent with our view that gradient variance is relative with the balance ration $r$ between regularization term and optimization term. Therefore, it inspires us to assume the relationship between the unknown balance ration $r$ with the easily available gradient variance $\tau$.

Moreover, there is a sudden decline of $r$ between 0.6 and 0.7 in Figure 1(a), which was not observed in other datasets or model architectures. This suggests that the behavior observed in Figure 1(a) is likely specific to the characteristics of the dataset or the model architecture used in that particular experiment. Additionally, considering that deep learning models can exhibit variability due to stochastic elements (e.g., initialization, sampling, or optimization), this decline might also be attributed to a random fluctuation specific to this experiment.

### B.3 EXPERIMENT DETAIL.

In the experiment of Figure 5, we set up 12 clients, and the dataset we used was CIFAR-10. To observe the data differences more intuitively, we introduced an extreme scenario of data heterogeneity: The local data of clients 1 to 4 contained only the first 5 classes of the CIFAR-10, clients 5 to 8 had only the left 5 classes, and clients 9 to 12 had local datasets that included all classes. The size

Table 10: Performance comparison when only adjust learning rate or weight decay.

| Dataset | Cifar10($\alpha = 0.1$) | Cifar10($\alpha = 100$) | Cifar100($\alpha = 0.1$) | Cifar100($\alpha = 100$) | Avg |
|---|---|---|---|---|---|
| FedLWS/only-reg | 61.54 | 74.92 | 34.39 | 38.89 | 52.44 |
| FedLWS/only-opt | 62.28 | 74.72 | 33.74 | 39.21 | 52.49 |
| FedLWS | **64.08** | **76.85** | **37.70** | **42.42** | **55.26** |

of the local dataset for each client was the same. We calculated the distances between the clients' local dataset via optimal transport, a methodology that has garnered widespread application within the realm of machine learning (Alvarez-Melis & Fusi, 2020; Gao et al., 2023; Ye et al., 2024), and the results are displayed in the left of Figure 5. Then, we conducted Federated Learning process, during which each client obtained its own local gradient after local training. We then compared the cosine distance between the local gradients, with the results displayed on the right of Figure 5. It can be seen that the relationships between the client vectors closely resemble those of the local data distributions. For example, client 1's client vector exhibits minimal differences with clients 2-4 due to their similar local data distributions. However, the differences between client 1 and clients 5-8 are much larger because of their highly divergent data distributions: client 1's local data contains only the first 5 classes, while clients 5-8 have only the last 5 classes. The differences between client 1 and clients 9-12 are smaller than those with clients 5-8, as clients 9-12 include data from all classes, making their distribution relatively closer to client 1. This demonstrates that the local gradient can effectively capture relevant information about the local data.

### B.4 ONLY ADJUST A SINGLE TERM.

As can be seen from the Equation 3, $\gamma$ influences both the optimization term and the regularization term. Our method calculates an appropriate value of $\gamma$ to balance the regularization and optimization terms during the training process. In other words, our method simultaneously considers both aspects to determine the optimal value of $\gamma$. To investigate the impact of our method on a single item, we conducted experiments as shown in Table 10, comparing two different variants: FedLWS/only-opt and FedLWS/only-reg. FedLWS/only-opt indicates adjusting only the optimization term during training, and FedLWS/only-reg indicates adjusting only the regularization term. The results indicate that adjusting only one of the terms does not perform well. This proves that our method effectively improves the model's performance by balancing the regularization and optimization terms during the training process.

### B.5 MORE EXPERIMENT OF LAYER-WISE WEIGHT SHRINKING

#### B.5.1 MODEL PARAMETERS

In Figure 8, we present the histogram of the final models' parameters when using CNN as the model architecture. Similar phenomena to those observed in Figure 3(b) of the main text can be observed. Layer-wise FedLWS drives a larger number of model parameters towards zero, thereby achieving an effect akin to weight decay and enhancing the model's generalization ability.

#### B.5.2 SHRINKING FACTOR FOR EACH LAYER.

In Figure 9, we illustrate the layer-wise shrinking factors calculated by our method for each layer during the training process across various model architectures. It can be observed that during the initial stages of training, there is a significant difference among the layers of the model, with the classifier exhibiting a more pronounced distinction compared to other layers. Previous works (Luo et al., 2021; Li et al., 2023b)) have demonstrated the specificity of the classifier in Federated Learning. Our observations in Figure 9 further validate this point. As the training progresses, the differences in the shrinking factors between layers gradually diminish. This indicates that our method primarily adjusts the aggregation process of the model during the early stages of training. As the training converges, the differences between clients gradually diminish, resulting in the calculated shrinking factors approaching 1.

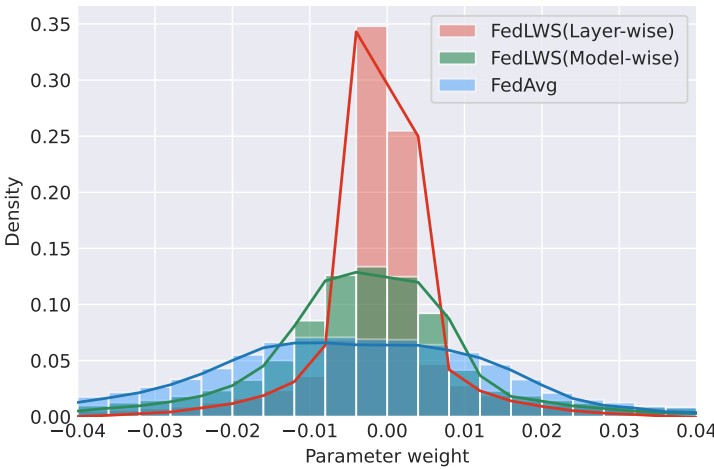

Figure 8: The histogram of final models' parameters. (CNN)

### B.5.3 COMPARE WITH FIXED LAYER-WISE SHRINKING FACTOR.

In Table 11, taking the FedAvg and FedDisco as the baselines, we compare the performance of our FedLWS and the method with fixed layer-wise $\gamma$ (the method in Figure 1(b)) under different datasets and degrees of heterogeneity. Fixed layer-wise $\gamma$ sets a fixed shrinking factor for each layer, and its value remains constant throughout the training process. Given that the CNN model comprises five layers, we set the $\gamma$ values between each consecutive layer to differ by 0.01. Therefore, in fixed layer-wise $\gamma$, the $\gamma$ of each layer is uniformly decreasing from 1 to 0.96. From Table 11, it can be observed that the performance of fixed layer-wise $\gamma$ shows a significant improvement in some scenarios, even surpassing FedLWS, indicating that considering weight shrinking in a layer-wise manner can indeed enhance the model's performance. However, its performance is not stable, and a significant deterioration is observed in certain situations (CIFAR-100, NIID($\alpha$=0.1)). Therefore, compared to fixed layer-wise $\gamma$, our approach is more robust. By dynamically adjusting the layer-wise shrinking factors in different scenarios, our method achieves better overall performance.

Table 11: Compare the performance of FedLWS and fixed layer-wise shrinking factor in both IID($\alpha$=100) and NIID($\alpha$=0.1) settings on CIFAR-10 and CIFAR-100.

| Dataset | CIFAR-10 | | CIFAR-100 | | |
|---|---|---|---|---|---|
| **Heterogeneity** | IID($\alpha$=100) | NIID($\alpha$=0.1) | IID($\alpha$=100) | NIID($\alpha$=0.1) | Mean Improvement |
| **Model** | CNN | CNN | CNN | CNN | |
| FedAvg | 70.19 | 59.66 | 32.25 | 30.18 | ——— |
| FedAvg+Fixed layer-wise $\gamma$ | **73.86**(↑ **3.67**) | 61.47(↑ 1.81) | **36.47**(↑ **4.22**) | 26.14(↓ 4.04) | 1.415 |
| FedAvg+FedLWS | 71.79(↑1.60) | **62.73**(↑**3.07**) | 33.24(↑0.99) | **32.51**(↑**2.33**) | **1.998** |
| FedDisco | 70.66 | 61.78 | 32.61 | 30.28 | ——— |
| FedDisco+Fixed layer-wise $\gamma$ | **73.83**(↑ **3.17**) | 62.37(↑ 0.59) | **36.48**(↑ **3.87**) | 25.73(↓ 4.55) | 0.770 |
| FedDisco+FedLWS | 71.36(↑0.70) | **64.79**(↑**3.01**) | 32.97(↑0.61) | **32.46**(↑**2.18**) | 1.625 |

### B.6 DIFFERENT LAYER TYPE.

In this section, we investigate the differences in shrinking factors calculated using our method across different types of layers. To ensure a fair and accurate comparison, we employed the MLP,

ResNet20, and ViT architectures for federated learning on the CIFAR-10 dataset. In the Table below, we presented the average shrinking factors for each layer type, where $\bar{\gamma}$ is the mean of layer-wise shrinking factors, e.g., MLP has 3 layers, corresponding to 3 layer-wise shrinking factors $\gamma_1, \gamma_1, \gamma_3$, and $\bar{\gamma} = \frac{(\gamma_1 + \gamma_2 + \gamma_3)}{3}$, ViT(Att.) represents the attention layers in ViT, and ViT(MLP) represents the MLP in ViT.

Table 12: Average shrinking factor for different types of layers.

|  | MLP | ResNet20 (Conv.) | ViT (Att.) | ViT (MLP) |
|---|---|---|---|---|
| $\bar{\gamma}$ | $0.980 \pm 0.011$ | $0.920 \pm 0.026$ | $0.995 \pm 0.003$ | $0.950 \pm 0.037$ |

It can be observed that the $\gamma$ values obtained for different types of layers vary significantly. This also demonstrates that our method can calculate the corresponding shrinking factors for different layer types. Notably, the shrinking factors for ViT(Att.) layers are closer to 1 and exhibit smaller differences across layers (low variance), indicating that weaker regularization is required. This can be attributed to the extensive parameter size of these layers, which minimizes the impact of gradient changes. Furthermore, in both MLP and ViT(MLP), the shrinking factor of the last layer is smaller than that of the other layers, e.g., the shrinking factors for the three MLP layers are 0.988, 0.984, and 0.968 respectively. This trend can be attributed to the fact that the gradient changes in the last layer of MLP are greater than those in the preceding layers, akin to the phenomenon of gradient vanishing. Consequently, the last layer requires stronger regularization (smaller shrinking factor). The experiments indicate that our method calculates the corresponding shrinking factor for different layer types, allowing for a more refined adjustment of the model aggregation process.

### B.7 TRAINING PROCESS

To evaluate the convergence of the proposed method, we examined the variation in test accuracy over rounds across multiple datasets and experimental configurations. Figure 10 illustrates the accuracy curves under various settings, including different datasets, heterogeneity degree $\alpha$, model architectures, client numbers, selection ratios, and local epochs E. As shown in Figure 10, our method has a similar convergence speed to FedAvg, the accuracy consistently increases with the number of rounds across all datasets, eventually reaching a stable plateau. This trend demonstrates the robustness and convergence of the proposed method under these configurations. In most cases, the accuracy exhibits rapid improvement during the initial stages of training, followed by a gradual stabilization as the model approaches convergence. Furthermore, the results reveal several insights into the impact of different configurations on convergence speed and final performance:

**Impact of $\alpha$:** Larger values of $\alpha$ result in smoother optimization processes, whereas smaller values may lead to more oscillations in accuracy during training. However, the method ultimately converges in both cases.

**Selection Ratios:** A smaller selection ratio results in slower improvement and more oscillations in training accuracy. Nonetheless, the overall trend remains stable, indicating the method's adaptability to varying levels of client participation.

**Local Training Epochs:** Increasing the number of local training epochs significantly accelerates global convergence, highlighting the importance of local updates in enhancing global optimization efficiency.

These observations collectively demonstrate the convergence properties of the proposed method under diverse experimental settings. The consistent upward trend in training accuracy and eventual stabilization across all scenarios confirm the effectiveness and robustness of the method in federated learning.

## C IMPLEMENTATION DETAILS

In this section, we provide details of the pseudo-code, experiments details, environment, datasets, model architectures, and hyperparameters of the experiment. We include the source code for implementing FedLWS in the Supplementary Material.

---

**Algorithm 1** FedLWS: Federated Learning with Adaptive Layer-wise Weight Shrinking

---

1: **Input:** Communication round $T$, local epoch $E$, local datasets $\{\mathcal{D}_1, ..., \mathcal{D}_K\}$, initial global model $\mathbf{w}_g^0 = \mathbf{w}_{gL}^0 \circ ... \circ \mathbf{w}_{g2}^0 \circ \mathbf{w}_{g1}^0$, where $\mathbf{w}_{gl}^0$ is the $l$-th layer of $\mathbf{w}_g^0$, combine with FL method $M$;

2: **Output:** Final global model $\mathbf{w}_g^T$;

3: **for** $t = 0$ **to** $T$ **do**

4:     Server sends global model $\mathbf{w}_g^t$ to each client;

    # Clients execute:

5:     **for** each client $k \in [K]$ **do**

6:         **If** $M = $ FedAvg **then**: $\mathbf{w}_k^t \leftarrow$ ClientUpdate$(\mathbf{w}_g^t, \mathcal{D}_k, E)$;

7:         **If** $M = $ FedProx **then**: $\mathbf{w}_k^t \leftarrow$ ClientUpdatewith$(\mathbf{w}_g^t, \mathcal{D}_k, E)$;# Use the Loss of FedProx

8:         Send $\mathbf{w}_k^t$ to server;

9:     **end for**

    # Server executes:

10:    Server aggregates the received model to generate the global model:

11:         $\widehat{\mathbf{w}}_g^{t+1} = \sum_{k=1}^K \lambda_k \mathbf{w}_k^t$ , $\lambda_k = \frac{\mathcal{D}_k}{\sum_{i=1}^K \mathcal{D}_i}$; # If $M$ adjusts aggregation weights, $\lambda_k$ is calculated using $M$.

> **Additional Step of our FedLWS**
>
> 12:    Computes the value of $\mathbf{g}_{kl}^t = \mathbf{w}_{kl}^t - \mathbf{w}_{gl}^t$, and $\tau_l^t = \frac{1}{K}\sum_{k=1}^K \|\mathbf{g}_{kl}^t - \mathbf{g}_{meanl}^t\|$;
>
> 13:    Computes the value of $\gamma_l^t = \frac{\|\mathbf{w}_{gl}^t\|}{\beta \tau_l^t \|\eta_g^t \mathbf{g}_{gl}^t\| + \|\mathbf{w}_{gl}^t\|}$;
>
> 14:    Server conducts layer-wise weight shrinking to obtain the updated global model $\mathbf{w}_g^{t+1} = \gamma_L^t(\widehat{\mathbf{w}}_{gL}^{t+1}) \circ ... \circ \gamma_2^t(\widehat{\mathbf{w}}_{g2}^{t+1}) \circ \gamma_1^t(\widehat{\mathbf{w}}_{g1}^{t+1})$;

15: **end for**

---

## C.1 PSEUDO-CODE

To facilitate a clearer understanding of our methodology, we present the pseudo-code of FedLWS in Algorithm 1, thereby providing a more intuitive representation of our method.

## C.2 ENVIRONMENT

We conduct experiments under Python 3.7.16 and Pytorch 1.13.1 (Paszke et al., 2019).

## C.3 DATASETS

In the experiment, we utilized four image classification datasets: CIFAR-10 (Krizhevsky et al., 2009), CIFAR-100 (Krizhevsky et al., 2009), FashionMNIST (Xiao et al., 2017), and Tiny-ImageNet (Chrabaszcz et al., 2017), which have been widely employed in prior Federated Learning methods (Li et al., 2023a; Ye et al., 2023; Luo et al., 2021). All these datasets are readily available for download online. To generate a non-IID data partition among clients, we employed Dirichlet distribution sampling $Dir_\alpha$ in the training set of each dataset, the smaller the value of $\alpha$, the greater the non-IID. In our implementation, apart from clients having different class distributions, clients also have different dataset sizes, which we believe reflects a more realistic partition in practical scenarios. We set $\alpha =$ 0.1, 0.5, and 100, respectively. When $\alpha$ is set to 100, we consider the data to be distributed in an IID manner. The data distribution across categories and clients is illustrated in Figure 11. Due to the large number of categories, we did not display the data distribution of CIFAR-100 and Tiny-ImageNet. Their distributions are similar to the other two datasets.

## C.4 HYPERPARAMETERS

If not mentioned otherwise, The number of clients, participation ratio, and local epoch are set to 20, 1, and 1, respectively. We set $\beta = 0.1$ for CNN models and $\beta = 0.01$ for ResNet models. We set the initial learning rates as 0.08 and set a decaying LR scheduler in all experiments; that is, in each round,

the local learning rate is 0.99*(the learning rate of the last round). We adopt local weight decay in all experiments. We set the weight decay factor as 5e-4. We use SGD optimizer as the clients' local optimizer and set momentum as 0.9.

### C.5 MODELS

For each dataset, all methods are evaluated with the same model architectures for a fair comparison. In Table 1 and 2, We use a simple CNN for FashionMNIST, ResNet20 (He et al., 2016) for CIFAR-10 and CIFAR-100, ResNet18 for Tiny-ImageNet, and TextCNN (Zhang & Wallace, 2015) for AG News.

**SimpleCNN.** The SimpleCNN is a convolution neural network model with ReLU activations. In this paper CNN consists of 3 convolutional layers followed by 2 fully connected layers. The first convolutional layer is of size (3, 32, 3) followed by a max pooling layer of size (2, 2). The second and third convolutional layers are of sizes (32, 64, 3) and (64, 64, 3), respectively. The last two connected layers are of sizes (64*4*4, 64) and (64, num_classes), respectively.

**ResNet, WRN, DenseNet and ViT.** We followed the model architectures used in (Li et al., 2023a; Dosovitskiy et al., 2020; Li et al., 2018). The numbers of the model names mean the number of layers of the models. Naturally, the larger number indicates a deeper network. For the Wide-ResNet56-4 (WRN56_4) in Table 5, "4" refers to four times as many filters per layer.

## D  THEORY ANALYSIS

In the following analysis, we omit $t$ to assist clarity. In Federated Learning, the global model is trained by aggregating client updates. The generalization gap is defined as:

$$\mathcal{G} = \left| \mathbb{E}_{x \sim \mathcal{D}}[L(\mathbf{w}; x)] - \frac{1}{K} \sum_{k=1}^{K} \mathbb{E}_{x \sim \mathcal{D}_k}[L(\mathbf{w}_k; x)] \right|,$$

where:

- $\mathbb{E}_{x \sim \mathcal{D}}[L(\mathbf{w}; x)]$ is the true risk (expected loss over the global distribution),
- $\frac{1}{K} \sum_{k=1}^{K} \mathbb{E}_{x \sim \mathcal{D}_k}[L(\mathbf{w}_k; x)]$ is the empirical risk (average loss over participating clients).

**Lemma D.1.** *Assume that the loss function $L(w)$ is Lipschitz-smooth and differentiable and its higher-order terms in the Taylor expansion are negligible. Then, the variance of client losses is proportional to the variance of client gradients:*

$$\mathbb{E}[(L(\mathbf{w}_k; \mathcal{D}_k) - L(\mathbf{w}; \mathcal{D}))^2] \propto \mathbb{E}[\|g_k - g_{mean}\|^2].$$

*Proof.* We first approximate the client loss $L(\mathbf{w}_k; \mathcal{D}_k)$ using a first-order Taylor expansion around the global model $\mathbf{w}$:

$$L(\mathbf{w}_k; \mathcal{D}_k) \approx L(\mathbf{w}; \mathcal{D}) + \nabla_{\mathbf{w}} L(\mathbf{w}; \mathcal{D})^\top (\mathbf{w} - \mathbf{w}_k),$$

where $\mathbf{w}_k$ is the locally updated model on client $k$, and $\mathbf{w} - \mathbf{w}_k \propto g_k$. We have:

$$(L(\mathbf{w}_k; \mathcal{D}_k) - L(\mathbf{w}; \mathcal{D}))^2 \approx (\nabla_{\mathbf{w}} L(\mathbf{w}; \mathcal{D})^\top (\eta g_k))^2.$$

Let us denote $\delta_k = g_k - g_{\text{mean}}$ and then we have:

$$(L(\mathbf{w}; \mathcal{D}_k) - L(\mathbf{w}; \mathcal{D}))^2 \propto (\nabla_{\mathbf{w}} L(\mathbf{w}; \mathcal{D})^\top \delta_k)^2.$$

Taking the expectation over all clients, we derive:

$$\mathbb{E}[(L(\mathbf{w}; \mathcal{D}_k) - L(\mathbf{w}; \mathcal{D}))^2] \propto \mathbb{E}[\|g_k - g_{\text{mean}}\|^2].$$

$\square$

**Theorem D.2.** *The following upper bound of the generalization gap exists:*

$$\mathcal{G} \leq C \cdot \sqrt{\frac{\tau}{K}}. \tag{10}$$

*Proof.* Using Bernstein's inequality, the probability of $\mathcal{G}$ is bounded as:

$$\Pr\left(\mathcal{G} \geq \epsilon\right) \leq 2 \exp\left(-\frac{K\epsilon^2}{2(\sigma^2 + M\epsilon/3)}\right),$$

where $\sigma^2 = \frac{1}{K}\sum_{k=1}^{K}\sigma_k^2$ is the variance of client losses and $M$ is the maximum deviation of individual client losses.

The expected generalization gap is bounded by integrating over $\epsilon$:

$$\mathbb{E}[\mathcal{G}] = \int_0^\infty \Pr(|\hat{R}(\mathbf{w}) - R(\mathbf{w})| \geq \epsilon)\, d\epsilon,$$

which can be further written as:

$$\mathbb{E}[\mathcal{G}] \leq \sqrt{\frac{2\sigma^2}{K}}.$$

Given Lemma D.1 and $\tau = \frac{1}{K}\sum_{k=1}^{K}\|g_k - g_{\text{mean}}\|$, the variance of client losses can be approximated by the variance of client gradients, i.e., $\sigma^2 \approx \tau$ Therefore, we have:

$$\mathbb{E}[\mathcal{G}] \leq \sqrt{\frac{2\tau}{K}}.$$

$\square$

**Discussion:** The above variance-based generalization bound provides a theoretical foundation for managing client heterogeneity in FL. It suggests that when $\tau$ is large, meaning that the client gradient variance is high, which also indicates that there is a heterogeneity between the clients. In this case, the generalization gap is increased as well. According to Eq. (7) in our main paper, higher $\tau$ leads to lower $\gamma$, which increases the strength of the regularization (Eq. (3) in our main paper). To summarize, our adaptive approach dynamically adjusts the strength of the regularization according to the current heterogeneity between the clients during training, which reduces the generalization gap in FL models.

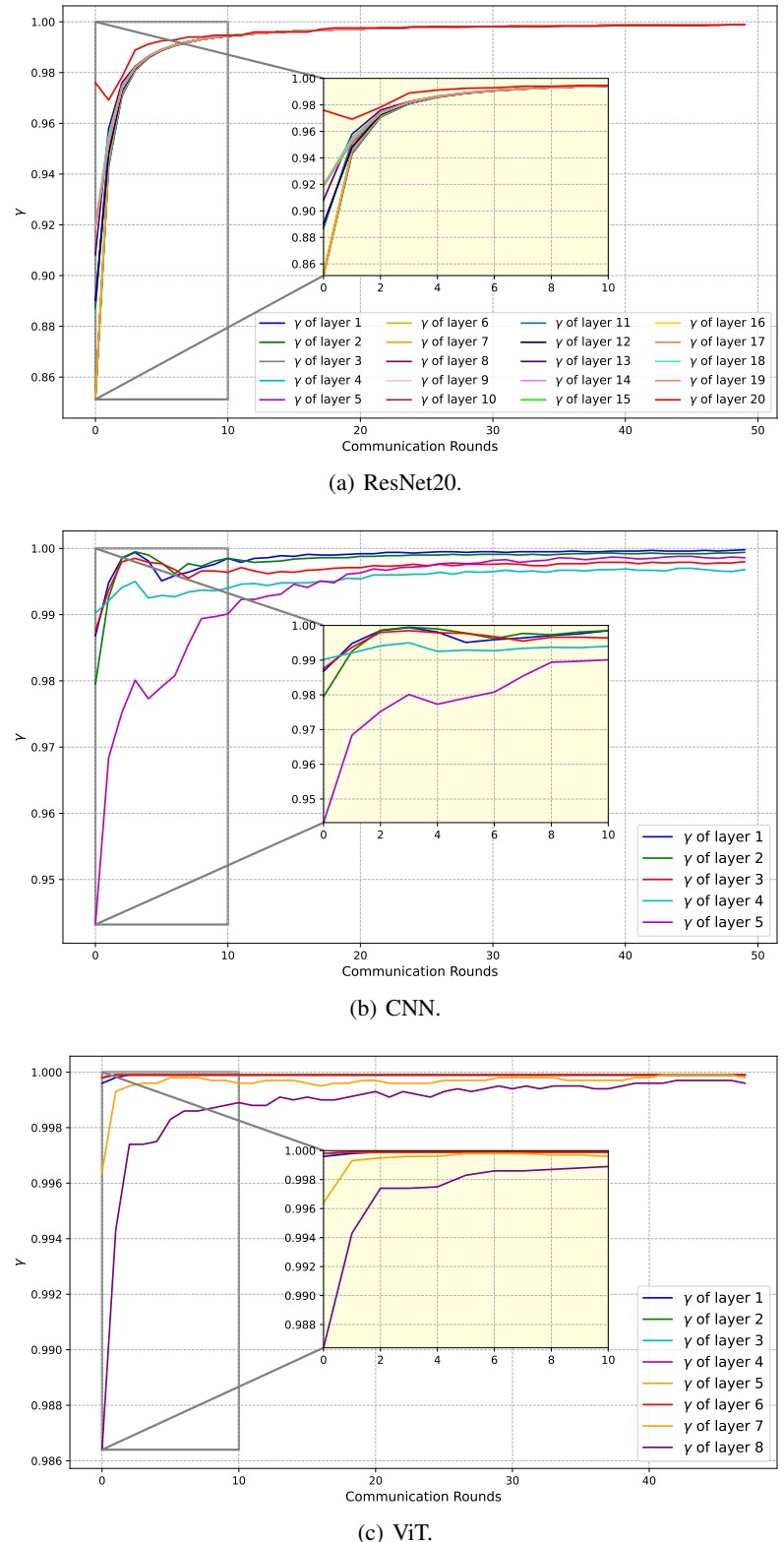

(a) ResNet20.

(b) CNN.

(c) ViT.

Figure 9: The value of each layer's shrinking factor for different model architectures during the training process, with the dataset being CIFAR-10.

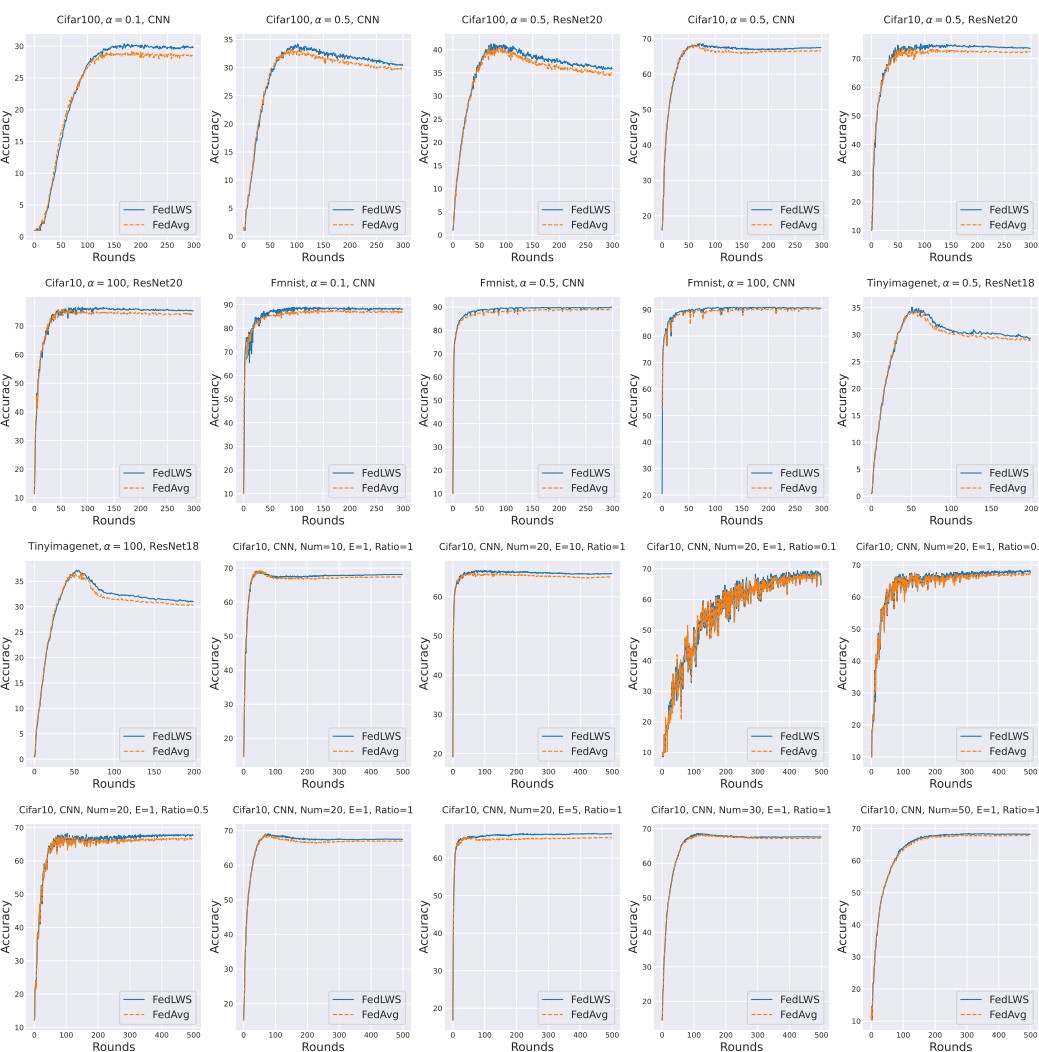

Figure 10: Training Accuracy Across Various Datasets and Experimental Configurations (client number, local epoch E, and select ratio).

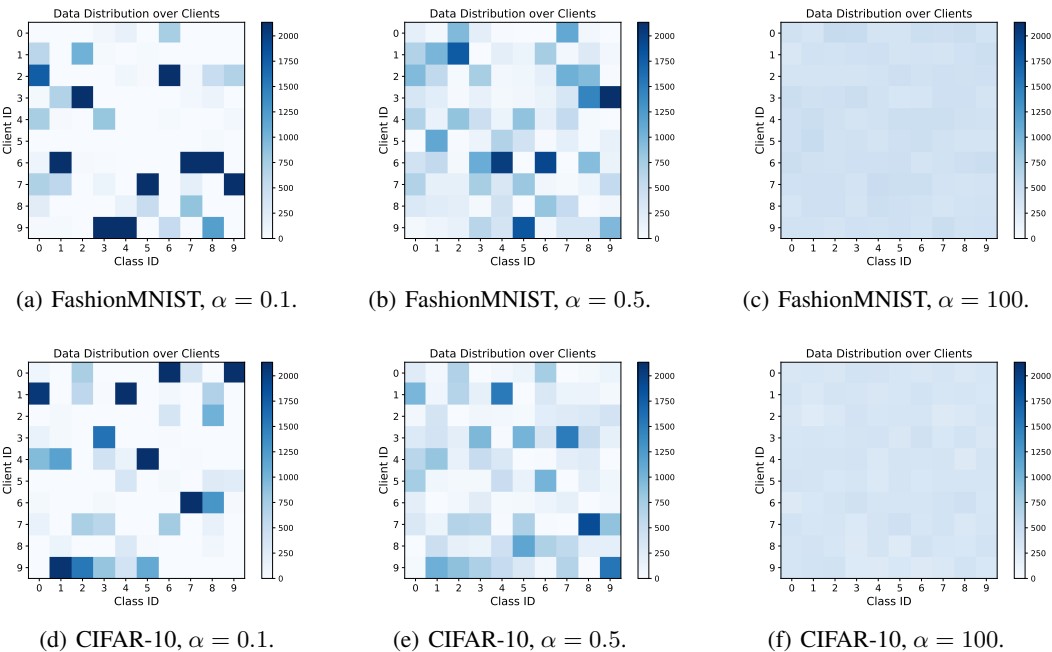

(a) FashionMNIST, $\alpha = 0.1$.  (b) FashionMNIST, $\alpha = 0.5$.  (c) FashionMNIST, $\alpha = 100$.

(d) CIFAR-10, $\alpha = 0.1$.  (e) CIFAR-10, $\alpha = 0.5$.  (f) CIFAR-10, $\alpha = 100$.

Figure 11: Data distribution over categories and clients.

