# OpenReview forum: "FedLWS: Federated Learning with Adaptive Layer-wise Weight Shrinking"
_ICLR.cc/2025/Conference — ICLR 2025 Poster_

### Official Review · Reviewer_motB · 2024-10-25

**Soundness:** 3
**Presentation:** 2
**Contribution:** 3
**Rating:** 6
**Confidence:** 4

**Summary:**

This paper introduces a method, FedLWS, to overcome the limitations of previous research named FedLAW, which leverages a Global Weight Shrinking (GWS) effect using a learnable shrinking factor based on proxy data. The authors identify two key issues with FedLAW: privacy concerns and considering the divergence across different layers of a deep neural network. To address these issues, they propose an alternative approach for calculating the shrinking factor using the variance of local model gradients. This method is based on the empirically observed relationship between the ratio of the regularization term to the optimization term and local gradient variance. Furthermore, this calculation enables the authors to propose a layer-wise shrinking factor for each layer, considering inter-layer differences for improved model generalization. Extensive experiments, including ablation studies, are conducted across various scenarios to verify the effectiveness of the proposed method. Overall, the authors present an effective alternative to address the limitations of previous research, and the experimental results appear to support its effectiveness. However, a more detailed analysis of the experimental results is necessary.

**Strengths:**

1. By proposing an alternative method for calculating the weight shrinking factor, the authors improve performance over the previous algorithm, reduce computation time on the server side, and eliminate privacy risks associated with the use of proxy data.

2. The authors analyze the characteristics of weight updates resulting from the application of the weight shrinking factor in the model agregation process of FL, validating these characteristics through equations and experiments.

3. The effectiveness of the proposed method is validated through extensive experiments conducted in various environments, as well as ablation studies under diverse conditions.

**Weaknesses:**

1. The validation of the proposed method’s effectiveness across various experimental conditions is a strong point; however, the results are presented as simple listings or repetitions of earlier sections without in-depth analysis, making it difficult to gain insights into how the proposed method operates from a representation perspective. For example, in lines 388–389, the authors describe the results in Table 1 by comparing them with the baseline method, FedLAW. The authors note that in some settings, the method underperforms compared to FedLAW, attributing this to the latter’s use of proxy data. However, as FedLAW utilizes proxy data in all cases, this explanation does not fully account for the variations in performance improvement across different cases.

2. There are several typos across multiple lines, and sentences with similar meanings are repeated, leading to a lack of coherence in the overall narrative.

**Questions:**

1. Is it correct to use the equal sign between the left and right sides of equation (4)? Would it be more appropriate to use a proportionality sign instead?

2. What does the sudden decline in the value in the range of 0.6 - 0.7 in Fig 1(a) signify?

3. The calculation of the layer-wise shrinking factor seems to increase the computational load, yet Table 3 reports only a 0.02-second increase. Is there a technique applied to compensate for the increase in computational load?

4. In Figure 3(a), why does the value of layer-wise gamma increase monotonically from early to later layers?

5. In section 5.3, the results in Table 4 and the narrative are inconsistent. While the performance has improved model-wise compared to FedAvg, it is stated otherwise. The author insists(line 448): on the CIFAR-10 dataset with NIID (α= 0.1), the performance of model-wise weight shrinking is not as effective as FedAvg, whereas the layer-wise weight shrinking effect outperforms FedAvg.

6. In Table 5, is the result for FedLAW with WRN56_4, indicated as 18.44, correct? If so, why is the accuracy for this setting significantly lower than that of the others?

7. In Fig 6, it seems to be no effect of beta. What is the reason for selecting beta as a hyperparameter?

---

> ### Author Response · Authors · 2024-11-24
> **Reply to Reviewer motB (Part 1)**
>
> **W1:** The experimental results are presented as simple listings without in-depth analysis.
>
> > **Response to W1:** Thank you for your valuable suggestions. Following your feedback, we have provided a more in-depth analysis in lines [386-399] of the revised manuscript, summarized below for your convenience. In our experiments, we set the proxy dataset for FedLAW to contain 200 samples, which aligns with the original settings in FedLAW [a]. For CIFAR-10, this proxy dataset includes 20 samples per class, providing sufficient information to optimize the shrinking factor effectively. However, for datasets like CIFAR-100 and TinyImageNet, the proxy dataset contains only 2 and 1 sample per class, respectively. This limited representation makes it difficult for FedLAW to train an optimal shrinking factor, which may explain the variations in its performance across different scenarios. Thank you again for pointing this out, which promotes us to refine the discussion and analysis.
> >
> > ---
> >
> > [a] Li, Zexi, et al. "Revisiting weighted aggregation in federated learning with neural networks." International Conference on Machine Learning. PMLR, 2023.
>
> ---
>
> **W2:** There are several typos across multiple lines, and sentences with similar meanings are repeated.
>
> > **Response to W2:** Thank you for pointing this out. We appreciate your careful reading of our manuscript and your feedback regarding typos and repetitive sentences. We have thoroughly reviewed the text and corrected all identified typos. Additionally, we have revised sections with repeated or redundant sentences to ensure a more concise and coherent narrative throughout the manuscript.
>
> ---
>
> **Q1:** Is it correct to use the equal sign between the left and right sides of equation (4)? Would it be more appropriate to use a proportionality sign instead?
>
> > **Response to Q1:** Thank you for pointing out this concern regarding the use of the equal sign in equation (4). We appreciate your careful review. We agree that the proportionality sign is more appropriate in this context, as it better reflects the relationship between the left-hand side and right-hand side of the equation. We have updated the equation accordingly in the revised manuscript and adjusted the accompanying explanation to ensure consistency and clarity. We believe this change enhances the accuracy and precision of the presentation. Thank you again for highlighting this point and helping us improve the manuscript.
>
> ---
>
> **Q2:** What does the sudden decline in the value in the range of 0.6 - 0.7 in Fig 1(a) signify?
>
> > **Response to Q2:** Thank you for your insightful question regarding the sudden decline in the range of 0.6–0.7 in Figure. 1(a), where we conduct the experiment on CIFAR10 with CNN as the backbone. To validate whether this phenomenon is present on the other datasets and model architectures, we further visualize the gradient variance $\tau$ and the balance ratio $r$ under different degrees of data heterogeneity across different datasets and model architectures. The results and discussions are added in **Appendix B.2. Figure 7** of our revised version.
>
> > We can see that there has been no sudden decline in other scenarios. This suggests that the observed behavior in Figure. 1(a) is likely specific to the characteristics of the dataset and the backbone used in that particular experiment.  One possible explanation is that the dataset contains certain features or distributions that make the model particularly sensitive within this range of values. In addition, given that deep learning models can exhibit variability due to stochastic elements (e.g., initialization, sampling, or optimization), the decline might also arise as a random fluctuation specific to this particular experiment. Besides, given these results in other scenarios, we can still observe that both $r$ and $\tau$ exhibit a corresponding reduction as the degree of data heterogeneity diminishes. It is also consistent with our view that gradient variance is very relative with the balance ration $r$ between regularization term and optimization term. Therefore, it inspires us to assume the relationship between the unknown balance ration $r$ with the easily available gradient variance $\tau$.
>
> > Thank you again for raising this question, which has helped us refine our analysis.

---

> ### Author Response · Authors · 2024-11-24
> **Reply to Reviewer motB (Part 2)**
>
> **Q3:** The calculation of the layer-wise shrinking factor seems to increase the computational load.
>
> > **Response to Q3:** Thank you for your thoughtful comment. We would like to clarify that our method only requires simple calculations instead of additional training or optimization steps. To provide a more intuitive illustration of the computational requirements of our method, in **Algorithm 1 in the Appendix**, we have highlighted the additional computations required by our method compared to FedAvg. Specifically, when the clients upload the local models $w_k$ to the server, we only need to compute:
> >
> > - The local gradient $g_{kl}^t = w_{kl}^t - w_{gl}^t$,
> > - The gradient variance $\tau^t_l = \frac{1}{K}\sum_{k=1}^{K}\|\|g_{kl}^t-g_{meanl}^t\|\|$, and
> > - The shrinking factor $\gamma^t_l = \frac{\|\|w_{gl}^t\|}{\beta \tau^t_l \|\|\eta_g^t g_{gl}^t\| + \|\|w_{gl}^t\|\|}$.
> >
> > Therefore, our approach only requires simple calculations, eliminating the need for optimization. As reported in original Table 3, the 0.02-second increase corresponds to experiments with the ResNet20 model. To provide a broader perspective, we have also conducted experiments with other model architectures. The results are included in **Table 3** and summarized below:
> >
> > | **Method**        | **CNN**   | **ResNet20** | **ViT**  | **WRN56_4** | **DenseNet121** |
> > | ----------------- | --------- | ------------ | -------- | ----------- | --------------- |
> > | **FedAvg**        | 0.019     | 0.10         | 0.18     | 0.561       | 1.359           |
> > | **FedLAW**        | 4.830     | 7.11         | 9.80     | 20.08       | 27.25           |
> > | **FedLWS (Ours)** | **0.035** | **0.12**     | **0.21** | **0.832**   | **1.756**       |
> >
> > It can be observed that while larger models may result in a slightly higher computational load, our method remains significantly more efficient compared to optimization-based approaches. We have included this analysis and the results of these additional experiments in the revised manuscript to better illustrate the efficiency of FedLWS. Thank you for highlighting this aspect, which allowed us to emphasize the efficiency of FedLWS further.
>
> ---
>
> **Q4:** In Figure 3(a), why does the value of layer-wise gamma increase monotonically from early to later layers?
>
> > **Response to Q4:** In Figure 3(a), we illustrate the variation of inter-layer differences across communication rounds on CIFAR-10 with ResNet20, where we only show the mean of layer-wise $\gamma$ of all layers (blue), $\gamma$ at the first layer (lilac), and $\gamma$ at the last layer (green) due to the limited space. The variation of layer-wise $\gamma$ for each layer is deferred to Figure 9(a) in the Appendix. As shown in Figure 9(a), we can find that the value of layer-wise $\gamma$ from early layers to later layers is not monotonically increasing. Besides, we also provide additional visualization of layer-wise $\gamma$ with CNN and ViT, shown in **Figure 9(b) and 9(c)**.
> >
> > It can be observed that during the initial stages of training, there is a significant difference among the layers of the model, with the classifier exhibiting a more pronounced distinction compared to other layers. This observation aligns with prior studies [a, b], which have highlighted the unique role and specificity of the classifier in Federated Learning. As depicted in Figure 9, our findings further validate this point. As the training progresses, the differences in the shrinking factors between layers gradually diminish. This indicates that our method primarily adjusts the aggregation process of the model during the early stages of training. As the training converges, the differences between clients gradually diminish, and the calculated shrinking factors approach 1. From this, it can be inferred that layer-wise $\gamma$ facilitates a more effective utilization of the global weight shrinking phenomenon and enhances model generalization by assigning optimal $\gamma$ values to each layer of the global model.
> >
> > ---
> >
> > [a] Luo, Mi, et al. "No fear of heterogeneity: Classifier calibration for federated learning with non-iid data." Advances in Neural Information Processing Systems 34 (2021): 5972-5984.
> > [b] Li, Zexi, et al. "No fear of classifier biases: Neural collapse inspired federated learning with synthetic and fixed classifier." Proceedings of the IEEE/CVF International Conference on Computer Vision. 2023.

---

> ### Author Response · Authors · 2024-11-24
> **Reply to Reviewer motB (Part 3)**
>
> **Q5:** The results in Table 4 and the narrative are inconsistent.
>
> > **Response to Q5:** We sincerely thank the reviewer for pointing out this inconsistency between Table 4 and the narrative in Section 5.3. This inconsistency is caused by an oversight of algorithmic improvements. We have updated the manuscript to correct this issue (lines 452-456). Thank you for your valuable feedback in helping us improve our manuscript.
>
> ---
>
> **Q6:** In Table 5, the result for FedLAW with WRN56\_4 is 18.44, why is the accuracy for this setting significantly lower than that of the others?
>
> > **Response to Q6:** In the original paper of FedLAW, the detailed parameter settings for the WRN56\_4 network structure were not provided. Therefore, we utilized their official code and the default parameter settings mentioned in their paper to conduct experiments, which might not be suitable for the WRN56\_4 network structure. Through our observations, we noticed that when using WRN56\_4 as the backbone, the $\gamma$ values learned by FedLAW were relatively small, which could affect the model's performance. After performing multiple sets of hyperparameter searches, we find that with a learning rate of 5e-3 and weight decay of 5e-5, the performance of FedLAW is improved to 36.6, which is still inferior to ours in revised Table 5. It indicates the robustness of our proposed FedLWS with different backbones. Furthermore, we revisited and re-trained the baseline methods that initially experienced training instabilities in the original experiments to ensure fairer comparisons. The updated results have been included in the manuscript and are highlighted for clarity.
>
> ---
>
> **Q7:** What is the reason for selecting beta as a hyperparameter?
>
> > **Response to Q7:** Thank you for your insightful comment. To clarify, the range of $\beta$ values presented in the original Figure 6 is relatively limited, as our goal is to highlight parameter settings most commonly used in practical applications. As shown in Eq. (7), $\beta$ controls the influence of $\tau$ on the shrinking factor $\gamma$. When $\beta$ approaches 0, $\gamma$ converges to 1, causing the model to degrade to the baseline. Conversely, when $\beta$ is too large, the calculated shrinking factor becomes excessively small, leading to model instability or failure. Our experiments indicate that a safe range for $\beta$ lies between 0.001 and 0.1. In response to your comment, we have updated the manuscript to expand on this explanation and included additional results in **Figure 6** to illustrate the effects of $\beta$ over a broader range. Thank you again for your valuable feedback, which has helped us improve the clarity and depth of our work.
>
> Overall, we hope that our responses can fully address your concerns and will be grateful for any feedback.

---

> > ### Comment · Reviewer_wdHM · 2024-11-30
> >
> > Thank you for your thoughtful and detailed responses to each of the comments. The response made to the manuscript, including the new theoretical analyses and experimental results in both the main text and the appendix, have addressed the majority of the issues I raised previously. These additions significantly enhance the robustness and clarity of the paper. However, there are still some errors and unresolved questions that need attention:
> >
> > - In the experiments conducted on the newly added text datasets, there appear to be errors in the experimental table. Specifically, why are there three rows labeled as "FedAvg"? Based on the experimental context, the fourth row seems to correspond to experiments involving FedProx.
> > - The inclusion of experiments involving CNN and ViT in Figure 9 is a valuable addition and helps extend the generalizability of FedLWS. However, I noticed that only the layer 20 of ResNet20 shows an unusual behavior where the γ value decreases initially and then increases—a pattern that appears counterintuitive. This phenomenon is not observed in other layers or architectures. Could the authors provide a clear explanation for this anomaly?
> > - In the revised version, the authors modified some of the experimental results from the original manuscript, explaining that these changes were due to adjustments in the experimental settings. However, this raises an important question: why did adjusting the hyperparameters result in changes to only some of the results? For example, in Table 5, the results for FedLAW on CIFAR-10 differ from those in the original manuscript, while the results for CIFAR-100 remain unchanged. Were different parameter settings used for these datasets? If so, this should be explicitly clarified. The authors should provide a clear and detailed description of the experimental environment and specific hyperparameter settings in the manuscript.
> >
> > In conclusion, while the manuscript represents a substantial improvement and successfully resolves many prior concerns, the above issues still need to be addressed. I appreciate the authors' efforts in revising the manuscript and look forward to seeing these points clarified and corrected.

---

> > ### Comment · Reviewer_motB · 2024-12-02
> >
> > Thank you for your detailed responses to each of the comments. After considering the rebuttal I have updated my score.

---

> > > ### Author Response · Authors · 2024-12-02
> > >
> > > Thank you for taking the time to review our response and for your positive feedback.

---

### Official Review · Reviewer_Ab6D · 2024-11-02

**Soundness:** 2
**Presentation:** 2
**Contribution:** 2
**Rating:** 6
**Confidence:** 4

**Summary:**

This work proposes a simple and practical model aggregation method to enhance the performance of various federated learning algorithms.

**Strengths:**

The proposed scheme is a direct intuitive scheme, which is easy to realize in practice. From the experimental results, the proposed algorithm can indeed play a role in improving performance.

**Weaknesses:**

Whether the proposed scheme is theoretically optimal is worth considering.

**Questions:**

1. The proposed weighting method used to determine the model aggregation accords with intuition but lacks theoretical proof. Can a theoretical proof be given to ensure that the weighting coefficient determined from the gradient variance perspective is optimal?

2. In Table 1, the performance of FedProx is generally worse than that of FedAvg. What is the reason for this phenomenon? The experiment should ensure that FedProx uses appropriate hyperparameters and that the comparison between the proposed LWA-enhanced methods and other baselines is fair, that is, the hyperparameters that make baselines' performance optimal should be used.

---

> ### Author Response · Authors · 2024-11-24
> **Reply to Reviewer Ab6D (Part 1)**
>
> **W1 & Q1:** Can a theoretical proof be given to ensure that the weighting coefficient determined from the gradient variance perspective is optimal?
>
> > **Response to W1&Q1:**  Thank you for your valuable suggestions.
> > Based on the reviewers’ feedback, we have included a variance-based generalization bound in **Appendix D**, providing a theoretical foundation for managing client heterogeneity in federated learning (FL). Specifically, when gradient variance $\tau$ is high, it indicates significant heterogeneity among clients, which in turn leads to an increased generalization gap. Our approach incorporates an adaptive regularization mechanism that dynamically adjusts the regularization strength based on $\tau$, effectively mitigating the negative impact of client heterogeneity on the generalization gap.
>
> > Regarding the formulation of the proposed function, the gradient variance $\tau$ in Equation (4) reflects client heterogeneity, while the ratio in Equation (5) determines the regularization strength via $\tau$. According to Equation (7) in the main paper, higher $\tau$ leads to lower $\gamma$, which strengthens the regularization. This dynamic adjustment mechanism leverages the gradient variance to guide the regularization strength, enhancing stability and reducing the generalization gap in the presence of high heterogeneity.
>
> > In summary, we believe that our theoretical analysis and experimental results provide meaningful insights and support for the proposed hypothesis and formulation, demonstrating their potential effectiveness in the examined scenarios. We sincerely thank the reviewers for highlighting this important aspect.

---

> ### Author Response · Authors · 2024-11-24
> **Reply to Reviewer Ab6D (Part 2)**
>
> **Q2:** The performance of FedProx is generally worse than that of FedAvg. What is the reason for this phenomenon? The experiment should ensure that FedProx uses appropriate hyperparameters and that the comparison is fair, that is, the hyperparameters that make baselines' performance optimal should be used.
>
> > **Response to Q2:** Thank you for your insightful comment. In our experiments, we used the default value for the regularization parameter in FedProx, as recommended in its original paper. Other hyperparameters, such as learning rate, batch size, and the number of communication rounds, were kept consistent across all baselines to ensure a fair comparison. The observed performance difference for FedProx may be attributed to the specific characteristics of our datasets or the degree of data heterogeneity, which could differ from those in the original FedProx study. Similar observations have also been reported in prior works [a, b], highlighting that dataset and heterogeneity differences may affect FedProx’s performance.
>
> > To address your concern and ensure a robust evaluation, we conducted additional experiments where the hyperparameter in FedProx was carefully tuned to optimize its performance. We have updated the results in Table 1 in our revision, where part of them is also listed in the table below for your convenience.
> >
> > | **Dataset**            | **FashionMNIST ($\alpha$=100)** | **FashionMNIST ($\alpha$=0.5)** | **FashionMNIST ($\alpha$=0.1)** | **CIFAR-10 ($\alpha$=100)** | **CIFAR-10 ($\alpha$=0.5)** | **CIFAR-10 ($\alpha$=0.1)** | **CIFAR-100 ($\alpha$=100)** | **CIFAR-100 ($\alpha$=0.5)** | **CIFAR-100 ($\alpha$=0.1)** | **Tiny-ImageNet ($\alpha$=100)** | **Tiny-ImageNet ($\alpha$=0.5)** | **Tiny-ImageNet ($\alpha$=0.1)** | **Average** |
> > | ----- | -------------- | --------- | --------- | ------- | ---------- | ------- | ---------- | --------- | ----- | -------- | -------- | ------- | ----------- |
> > | **FedAvg**             | 90.44                           | 90.04                           | 88.62                           | 76.01                       | 74.47                       | 61.04    | 41.46        | 37.21      | 36.71          | 36.31            | 34.43                            | 29.44      | 58.02       |
> > | **FedAvg+LWS (Ours)**  | **90.99**                       | **90.33**                       | **88.99**                       | **76.85**                   | **75.63**                   | **64.08**   | **42.42**                    | **41.03**                    | **37.70**                    | **37.16**                        | **35.12**                        | **31.34**                        | **59.30**   |
> > | **FedProx (Tuned)**    | 91.24                           | 90.69                           | 88.78                           | 73.96                       | 73.27                       | 60.62                       | 38.15                        | 39.35                        | 34.60                        | 35.03                            | 34.32                            | 29.37                            | 57.45       |
> > | **FedProx+LWS (Ours)** | **91.35**                       | **91.24**                       | **89.25**                       | **74.34**                   | **74.55**                   | **62.54**                   | **38.64**                    | **39.93**                    | **35.37**                    | **35.29**                        | **34.98**                        | **30.68**                        | **58.18**   |
> >
> > While parameter tuning enhances the performance of FedProx, its average accuracy remains inferior to that of FedAvg. This is primarily because under conditions where $\alpha=100$ (i.e., local data is approximately IID), FedProx tends to perform worse than FedAvg. This is likely because FedProx incorporates regularization during client training to mitigate the challenges of data heterogeneity in federated learning. However, in scenarios where data distribution is nearly IID, such regularization may not provide a benefit and can even limit performance. Furthermore, the combination of FedProx with our method still shows an additional improvement, demonstrating the adaptability and effectiveness of our proposed FedLWS in enhancing existing FL methods. We appreciate your valuable feedback, which has allowed us to provide a more comprehensive explanation and additional results to strengthen our study.
> >
> > ---
> >
> > [a] Li, Zexi, et al. "Revisiting weighted aggregation in federated learning with neural networks." International Conference on Machine Learning. PMLR, 2023.
> >
> > [b] Zhang, Jianqing, et al. "Gpfl: Simultaneously learning global and personalized feature information for personalized federated learning." *Proceedings of the IEEE/CVF International Conference on Computer Vision*. 2023.
>
> Overall, we hope that our responses can fully address your concerns and will be grateful for any feedback.

---

> > ### Comment · Reviewer_Ab6D · 2024-12-01
> > **thanks for the response**
> >
> > The authors solved my doubts. However, since each algorithm involves a variety of hyperparameters, the comparison of algorithm performance may need to be more careful, and the default hyperparameter of the baseline algorithm may not be optimal on different datasets. Nevertheless, the authors' experimental results show that the proposed method can improve performance. Therefore, I keep my previous score unchanged.

---

> > > ### Author Response · Authors · 2024-12-02
> > >
> > > Thank you for taking the time to review our response and for your valuable feedback.

---

### Official Review · Reviewer_wdHM · 2024-11-03

**Soundness:** 3
**Presentation:** 3
**Contribution:** 2
**Rating:** 6
**Confidence:** 4

**Summary:**

This paper presents FedLWS (Federated Learning with Adaptive Layer-wise Weight Shrinking), a novel model aggregation strategy designed to improve the generalization of global models in Federated Learning (FL). Recognizing that a shrinking factor (sum of aggregation weights) less than 1 can enhance model generalization, FedLWS adaptively adjusts the shrinking factor at each layer based on gradient dynamics without requiring a proxy dataset, thereby reducing privacy risks. The approach calculates layer-specific shrinking factors during training, leveraging the unique characteristics of each layer, and is compatible with existing FL methods. Extensive experiments demonstrate FedLWS’s consistent performance improvements over state-of-the-art methods across diverse scenarios. A noted limitation is its current applicability only to scenarios where client models share the same architecture. Future work will address this limitation by exploring heterogeneous model scenarios.

**Strengths:**

This paper addresses privacy concerns raised in prior research regarding the use of proxy datasets, and adopts an intuitive assumption in federated learning to dynamically adjust the shrinking factor (γ) layer-by-layer based on gradient variance. This approach aims to enhance model generalization while preserving privacy. The paper is well-structured, with extensive experimental support provided in both the main text and appendices. The key strengths of this approach are as follows:

- FedLWS avoids the need for proxy datasets, reducing potential privacy risks and making it more suitable for sensitive applications.
- The layer-wise adjustment of the shrinking factor (γ) allows FedLWS to adapt to specific layer characteristics, resulting in better model generalization across diverse and heterogeneous datasets.
- As a “plug-and-play” approach, FedLWS can be easily integrated with existing federated learning methods, improving their performance without requiring substantial modifications to the training process.

**Weaknesses:**

1.  The introduction does a good job of citing relevant works to contextualize the study. However, it might be beneficial to discuss more recent research on adaptive weight shrinkage and layer-wise adaptation in FL, if available. This could provide a clearer picture of how the proposed FedLWS method aligns with and advances the current research landscape in FL.
2.  This paper introduces a thoughtful approach by linking the shrinking factor to gradient variance, which allows dynamic adjustment of regularization. This concept appears promising and well-motivated, especially as it addresses practical deployment issues by eliminating the need for a proxy dataset. However, the authors could further clarify how this method stands out from other adaptive approaches. Are there any existing methods that also use gradient variance for similar adjustments, and if so, how does FedLWS provide additional benefits?
3. The paper presents an intuitive hypothesis on the relationship between regularization, optimization, and gradient variance. Additional theoretical support or references could help strengthen this hypothesis. While the authors propose a function to capture this relationship, further clarification on the rationale behind this formulation would be valuable. Could a theoretical proof or justification for this relationship be provided?
4. FedLWS employs a dynamically adjusted shrinking factor γ to balance regularization and optimization, but it is unclear whether this adjustment strategy could negatively impact convergence in federated learning. Frequent adjustments to the shrinking factor during multiple communication rounds may lead to instability or slower convergence. The authors could analyze the convergence behavior of FedLWS over multiple communication rounds, especially under varying data heterogeneity or client participation rates. Providing a convergence analysis or additional experimental results would help ensure that the method maintains stable convergence properties in federated learning.
5.  The modularity of FedLWS as a “plug-and-play” solution is a valuable contribution. However, the paper could benefit from a more detailed example of how FedLWS integrates with popular FL methods like FedAvg or FedProx. For instance, describing specific integration steps for one or two methods would help readers understand the practical applicability of FedLWS.
6. FedLWS employs a strategy of assigning different shrinking factors to different layers, based on the assumption that each layer requires a distinct level of regularization. However, in federated learning, different types of layers (e.g., convolutional layers and fully connected layers) may exhibit significantly different behaviors, and a simple layer-wise assignment may not be suitable for all types of layers. Could further experiments be conducted to verify the effectiveness of layer-wise shrinking across various types of layers? Additionally, it may be worth considering adjustments to the shrinking factor allocation based on the function or structural characteristics of each layer (e.g., differences between shallow and deep layers). This could potentially help FedLWS more effectively optimize each layer of the model.
7. The results in Tables 1 and 2 effectively demonstrate that FedLWS consistently improves accuracy across datasets and heterogeneity levels, especially for more challenging tasks. However, a more detailed explanation of why FedLWS performs differently across various datasets would be helpful. For instance, why does FedLWS perform better on datasets with higher complexity or heterogeneity? This is particularly noteworthy given that prior work had the advantage of an additional proxy dataset as prior knowledge.
8. In Figure 1(a), we observe a sudden drop when α  reaches 0.7, followed by a return to stability. Has the author considered the reasons behind this occurrence? In Figure 1(b), even a very small decrease (e.g., 0.01) has a significant impact on the results. Why is this the case?  The use of a 5-layer CNN model with uniformly decreasing values from 1 to 0.96 is an interesting experimental design choice. However, the authors could clarify the rationale for this specific range and pattern. Was the decrease from 1 to 0.96 based on empirical observation, theoretical insights, or a predetermined hypothesis? Including a brief justification for this choice could add depth to the experimental setup.
9. The calculation of layer-wise shrinking factors appears to significantly increase computational overhead, especially in deep or large models such as ViT or BERT. Could the authors provide an analysis of this aspect?
10.  Although FedLWS shows performance advantages in most experiments, it may underperform compared to FedLAW in certain cases. The authors attribute this entirely to FedLAW’s use of a proxy dataset, but this explanation seems insufficient, as FedLAW consistently uses a proxy dataset across all experimental settings. The authors should analyze under which experimental conditions FedLAW outperforms FedLWS and examine whether specific variables might explain this phenomenon.
11.  Although the paper includes multiple image and text datasets, it does not explain whether these datasets offer generalizability. In particular, AG News, as the sole text dataset, is insufficient to fully validate FedLWS’s performance on other types of text data or time-series data.
12.  Privacy is an essential consideration in FL, and the authors have addressed this well by removing the reliance on a proxy dataset. However, additional details on how FedLWS manages potential privacy risks, especially during gradient and parameter calculations, would be useful. Additionally, it would be valuable to discuss if FedLWS has any compatibility with existing privacy-preserving techniques like differential privacy.
13.  The comparison between FedLWS and adaptive weight decay methods (AWD and AdaDecay) is insightful, showing FedLWS’s advantage in FL. Adding a brief explanation of why FedLWS outperforms weight decay in the FL context (e.g., due to its layer-wise adaptability) would make this comparison more compelling. The authors do not explain this reason in Appendix A.2.

**Questions:**

- How does the proposed layer-wise dynamic adjustment method demonstrate superiority over traditional dynamic adjustment techniques in the context of weight aggregation?
- Can the authors provide a detailed background on prior experimental findings to help readers understand the motivation behind the approach?
- In the experimental section, why does the proposed method perform better, and can this be explained rather than just described?

---

> ### Author Response · Authors · 2024-11-24
> **Reply to Reviewer wdHM (Part 1)**
>
> **W1&Q1:** Discuss more recent research and related work and how does the proposed layer-wise dynamic adjustment method demonstrate superiority over traditional dynamic adjustment techniques?
>
> > **Response to W1&Q1:**  Thank you for your valuable feedback and insightful questions. In the revised manuscript, we have included discussions of several recent works that explore related concepts.
> >
> > **Comparison with Layer-Wise Adaptive Aggregation Techniques ([a], [b], [c]):** While our method shares the concept of layer-wise aggregation, there are key distinctions:
> >
> > - The method in [a] adjusts the aggregation frequency for each layer to reduce communication costs while accounting for inter-layer differences.
> > - The method in [b] focuses on personalized FL, it designs a hyper-network to predict the layer-wise aggregation weights for each client.
> >
> > - The method in [c] leverages the similarity between local and global models to dynamically determine the aggregation weights.
> >
> > - In contrast, our method is neither focused on aggregation frequency adjustment nor layer-wise aggregation weights adjustment. Instead, we propose an adaptive layer-wise weight shrinking step after model aggregation to mitigate aggregation bias, which is both computationally efficient and modular, enabling seamless integration with various FL frameworks and baselines.
> >
> > **Comparison with Weight Shrinking Techniques [d]:** FedLAW [d] identifies the global weight shrinking phenomenon and jointly learns the optimal shrinking factor $\gamma$ and aggregation weights $\lambda$ on the server. However, there are several challenges with this method:
> >
> > - Dependency on Proxy Dataset: FedLAW requires a proxy dataset with a distribution identical to the test dataset to optimize $\gamma$ and $\lambda$. This assumption is unrealistic in many real-world scenarios due to privacy concerns, as obtaining such a proxy dataset is often impractical.
> >
> > - Coupling of Shrinking Factor and Aggregation Weights: FedLAW jointly optimizes $\gamma$ and $\lambda$, making it challenging to combine with other aggregation strategies in FL.
> >
> > - Neglect of Layer-Wise Variations: FedLAW applies a uniform shrinking factor across all layers, ignoring the inter-layer differences in deep models that can be critical in non-IID scenarios.
> >
> >
> > To address these challenges, we propose our method FedLWS:
> >
> > - No Proxy Dataset Required: FedLWS directly calculates the shrinking factor using readily available gradients and global model parameters, avoiding the need for a proxy dataset.
> >
> > - Layer-Wise Design: Our method explicitly accounts for inter-layer differences by designing the shrinking factor in a layer-wise manner, making it more effective for handling heterogeneity in FL.
> >
> > - Decoupled Shrinking Factor and Aggregation Weights: FedLWS decouples these two aspects, enabling flexible integration with various FL aggregation strategies.
> >
> > - Lower Computational Cost: Unlike FedLAW, which requires optimization on the server, our method directly calculates the shrinking factor, significantly reducing computational overhead.
> >
> >
> > We hope these additions will help readers better understand how FedLWS contributes to the current advancements in FL. Thank you for your valuable feedback, which encouraged us to further refine the introduction and provide a clearer context for our work.
> >
> > ---
> >
> > [a] Lee, Sunwoo, Tuo Zhang, and A. Salman Avestimehr. "Layer-wise adaptive model aggregation for scalable federated learning." Proceedings of the AAAI Conference on Artificial Intelligence. Vol. 37. No. 7. 2023.
> >
> > [b] Ma, Xiaosong, et al. "Layer-wised model aggregation for personalized federated learning." Proceedings of the IEEE/CVF conference on computer vision and pattern recognition. 2022.
> >
> > [c] Rehman, Yasar Abbas Ur, et al. "L-dawa: Layer-wise divergence aware weight aggregation in federated self-supervised visual representation learning." Proceedings of the IEEE/CVF international conference on computer vision. 2023.
> >
> > [d] Li, Zexi, et al. "Revisiting weighted aggregation in federated learning with neural networks." International Conference on Machine Learning. PMLR, 2023.

---

> ### Author Response · Authors · 2024-11-24
> **Reply to Reviewer wdHM (Part 2)**
>
> **W2:** Are there any existing methods that also use gradient variance for similar adjustments, and if so, how does FedLWS provide additional benefits?
>
> > **Response to W2:** Thank you for your insightful comment and for recognizing the potential of our method. To the best of our knowledge, there are no existing methods that use gradient variance to compute model shrinking factors in a manner similar to our approach. However, several federated learning methods employ dynamic adjustments for model aggregation weights. For example: In [a], the similarity between local and global models is used to dynamically determine aggregation weights. In [b], a hypernetwork is designed to predict layer-wise aggregation weights for each client in the context of personalized federated learning.
>
> > Unlike these methods that dynamically adjust aggregation weights during the aggregation process, our approach performs weight shrinking on the aggregated global model after aggregation, effectively refining and adjusting the aggregation process. This approach provides a flexible and effective way to mitigate aggregation bias caused by client heterogeneity. Additionally, our method is compatible with many existing federated learning approaches, allowing for seamless integration.
>
> > Thank you for raising this important point, which allowed us to further elaborate on the novelty and versatility of FedLWS.
> >
> > ---
> >
> > [a] Rehman, Yasar Abbas Ur, et al. "L-dawa: Layer-wise divergence aware weight aggregation in federated self-supervised visual representation learning." Proceedings of the IEEE/CVF international conference on computer vision. 2023.
> >
> > [b] Ma, Xiaosong, et al. "Layer-wised model aggregation for personalized federated learning." Proceedings of the IEEE/CVF conference on computer vision and pattern recognition. 2022.
>
> **W3:** Could a theoretical proof or justification for  the relationship between regularization, optimization, and gradient variance be provided?
>
> > **Response to W3:**  We appreciate the reviewer’s valuable suggestions. Based on the reviewers’ feedback, we have included a variance-based generalization bound in **Appendix D**, providing a theoretical foundation for managing client heterogeneity in federated learning. Specifically, when gradient variance $\tau$ is high, it indicates significant heterogeneity among clients, which in turn leads to an increased generalization gap. Our approach incorporates an adaptive regularization mechanism that dynamically adjusts the regularization strength based on $\tau$, effectively mitigating the negative impact of client heterogeneity on the generalization gap.
>
> > Regarding the formulation of the proposed function, the gradient variance $\tau$ in Equation (5) reflects client heterogeneity, while the ratio in Equation (4) determines the regularization strength via $\tau$. According to Equation (7) in the main paper, higher $\tau$ leads to lower $\gamma$, which strengthens the regularization. This dynamic adjustment mechanism leverages the gradient variance to guide the regularization strength, enhancing stability and reducing the generalization gap in the presence of high heterogeneity.
>
> > In summary, our theoretical analysis and experimental results collectively provide substantial support for the proposed hypothesis and formulation, demonstrating their potential effectiveness in the examined scenarios. Thank you for highlighting this important aspect.

---

> ### Author Response · Authors · 2024-11-24
> **Reply to Reviewer wdHM (Part 3)**
>
> **W4:** The authors could analyze the convergence behavior of FedLWS over multiple communication rounds, especially under varying data heterogeneity or client participation rates.
>
> > **Response to W4:** Thank you for your valuable suggestions. To evaluate the convergence of our proposed FedLWS, we examined the variation in test accuracy over rounds under various settings, including different datasets, heterogeneity degree $\alpha$, model architectures, client numbers, selection ratios, and local epochs E. The result is shown in **Figure 10 in the Appendix**. It can be observed that our method has a similar convergence speed to FedAvg; the accuracy consistently increases with the number of rounds across all datasets, eventually reaching a stable plateau. This trend demonstrates the robustness and convergence of the proposed method under these configurations. In most cases, the accuracy exhibits rapid improvement during the initial stages of training, followed by a gradual stabilization as the model approaches convergence. Furthermore, the results reveal several insights into the impact of different configurations on convergence speed and final performance:
> >
> > - **Impact of $\alpha$:** Larger values of $\alpha$ result in smoother optimization processes, whereas smaller values may lead to more oscillations in accuracy during training. However, the method ultimately converges in both cases.
> >
> > - **Selection Ratios:** A smaller selection ratio results in slower improvement and more oscillations in training accuracy. Nonetheless, the overall trend remains stable, indicating the method's adaptability to varying levels of client participation.
> >
> > - **Local Training Epochs:** Increasing the number of local training epochs significantly accelerates global convergence, highlighting the importance of local updates in enhancing global optimization efficiency.
> >
> > These observations collectively demonstrate the convergence properties of the proposed method under diverse experimental settings. The consistent upward trend in accuracy and eventual stabilization across all scenarios confirm the effectiveness and robustness of our method.
>
> **W5:** More detailed example of how FedLWS integrates with popular FL methods.
>
> > **Response to W5:** Thank you for your valuable comment. Following your suggestion, we further show the workflow of our method in **Algorithm 1 in the Appendix**. In Algorithm 1, we have highlighted the additional steps required by our method to more intuitively illustrate how our method integrates with other approaches. When integrating FedLWS with FedAvg, FedProx, FedDisco, and other methods, the only modification needed is after the aggregation step. In these algorithms, model aggregation typically occurs at the end of each communication round. Our method introduces an additional weight shrinking step following the aggregation to adjust the model update process, where Equation (7) is used to compute the shrinking factors. These simple steps underscore the flexibility of FedLWS and its ease of adoption.

---

> ### Author Response · Authors · 2024-11-24
> **Reply to Reviewer wdHM (Part 4)**
>
> **W6:**  Could further experiments be conducted to verify the effectiveness of layer-wise shrinking across various types of layers?
>
> > **Response to W6:** Thank you for your valuable suggestion. We have conducted further experiments to investigate the differences in shrinking factors calculated by our method across various types of layers. To ensure a comprehensive and fair evaluation, we tested our method using MLP, ResNet20, and ViT architectures in a federated learning setting on the CIFAR-10 dataset. In the table below, we presented the average shrinking factors for each layer type, where $\bar \gamma$ is the mean of layer-wise shrinking factors, e.g., MLP has 3 layers, corresponding to 3 layer-wise shrinking factors $\gamma_1$, $\gamma_2$, $\gamma_3$, and $\bar \gamma=\frac{(\gamma_1+\gamma_2+\gamma_3)}{3}$, ViT(Att.) represents the attention layers in ViT, and ViT(MLP) represents the MLP in ViT.
>
> > |                | MLP               | ResNet20 (Conv.)  | ViT (Att.)        | ViT (MLP)         |
> > | -------------- | ----------------- | ----------------- | ----------------- | ----------------- |
> > | $\bar{\gamma}$ | $0.980 \pm 0.011$ | $0.920 \pm 0.026$ | $0.995 \pm 0.003$ | $0.950 \pm 0.037$ |
>
> > It can be observed that the $\gamma$ values obtained for different types of layers vary significantly. This also demonstrates that our method can calculate the corresponding shrinking factors for different layer types. Notably, the shrinking factors for ViT(Att.) layers are closer to 1 and exhibit smaller differences across layers (low variance), indicating that weaker regularization is required. This can be attributed to the extensive parameter size of these layers, which minimizes the impact of gradient changes.
> > Furthermore, in both MLP and ViT(MLP), the shrinking factor of the last layer is smaller than that of the other layers, e.g., the shrinking factors for the three MLP layers are 0.988, 0.984, and 0.968 respectively. This trend can be attributed to the fact that the gradient changes in the last layer of MLP are greater than those in the preceding layers, akin to the phenomenon of gradient vanishing. Consequently, the last layer requires stronger regularization (smaller shrinking factor). The experiments indicate that our method calculates the corresponding shrinking factor for different layer types, allowing for a more refined adjustment of the model aggregation process.
>
> **W7:** Why does FedLWS perform better on datasets with higher complexity or heterogeneity? This is noteworthy given that prior work had the advantage of prior knowledge.
>
> > **Response to W7:** Thank you for your insightful comment regarding the performance differences of FedLWS across various datasets. As demonstrated in many prior studies, FedAvg already achieves strong results on relatively simple tasks or datasets with consistent data distributions (i.e., IID settings). In such scenarios, the differences between client models are relatively small, and consequently, the $\gamma$ computed using Equation (7) in our method tends to be closer to 1. This means that FedLWS's behavior aligns more closely with FedAvg under these conditions. However, when dealing with datasets exhibiting higher complexity or heterogeneity (non-IID settings), the client model differences become more pronounced. In these cases, FedLWS's adaptive mechanism effectively mitigates these differences, leading to a more balanced and accurate global model. This is why the advantages of FedLWS are more evident in challenging tasks or under high heterogeneity.
>
> > Regarding methods that leverage prior knowledge, our method can naturally complement these approaches to further enhance model performance. For instance, as demonstrated in our work, combining FedLWS with FedDisco resulted in better performance than combining FedLWS with FedAvg. This highlights the potential synergy between our method and techniques that utilize prior knowledge.
>
> > We have included this explanation and clarification in the revised manuscript. Thank you for your valuable feedback.

---

> ### Author Response · Authors · 2024-11-24
> **Reply to Reviewer wdHM (Part 5)**
>
> **W8:** The concern about Figure 1(a) and Figure 1(b).
> > **Response to W8:** Thank you for your thoughtful observations and questions regarding Figures 1(a) and 1(b). We are happy to provide clarification on these points.
> > - **Regarding Figure 1(a):** To validate whether this phenomenon is present on the other datasets and model architectures, we further visualized the gradient variance $\tau$ and the balance ratio $r$ under different degrees of data heterogeneity across different datasets and model architectures. The results are presented in **Figure 7 in the Appendix** of our revised version. We can see that there has been no sudden decline in other scenarios. This suggests that the observed behavior in Figure 1(a) is likely specific to the characteristics of the dataset used in that particular experiment. One possible explanation is that the dataset contains certain features or distributions that make the model particularly sensitive within this range of values. In addition, given that deep learning models can exhibit variability due to stochastic elements (e.g., initialization, sampling, or optimization), the decline might also arise as a random fluctuation specific to this particular experiment.  Besides, given these results in other scenarios, we can still observe that both $r$ and $\tau$ exhibit a corresponding reduction as the degree of data heterogeneity diminishes. It is also consistent with our view that gradient variance is very relative with the balance ration $r$ between regularization term and optimization term. Therefore, it inspires us to assume the relationship between the unknown balance ration  $r$ with the easily available gradient variance $\tau$. Thank you again for raising this question, which has helped us refine our analysis.
> >
> > - **Regarding Figure 1(b):** The significant impact of even small decreases (e.g., 0.01) is because fixed shrinkage factors cannot adapt dynamically during training. This means their values influence the training process throughout iterations, amplifying the impact of even small changes. This is why the shrinkage factors set in Figure 1(b) are close to 1, ensuring minimal disruption to training stability. In contrast, our method dynamically adjusts the shrinkage factors during training, providing better adaptability and performance. The design of layer-wise $\gamma$ values uniformly decreasing from 1 to 0.96 in Figure 1(b) is based on the hypothesis that later layers in client models exhibit larger differences due to non-IID data distributions, requiring stronger regularization with smaller shrinking factors. As shown in **Figure 9(b) in the Appendix**, this trend is indeed observed in CNN models. However, it does not necessarily hold for other models. This observation further underscores the necessity of designing adaptive weight shrinking to accommodate varying behaviors across different architectures.
> >
> > Thank you again for raising this question, which has helped us refine our analysis.
>
> **W9:** Could the authors provide an analysis of the computational overhead?
>
> > **Response to W9:** Thank you for your thoughtful comment. We would like to clarify that our method only requires simple calculations instead of additional training or optimization steps. To provide a more intuitive illustration of the computational requirements of our method, in **Algorithm 1 in the Appendix**, we have highlighted the additional computations required by our method compared to FedAvg. Specifically, when the clients upload the local models $w_k$ to the server, we need to compute the local gradient $g_{kl}^t = w_{kl}^t - w_{gl}^t$, gradient variance $\tau^t_l$, and the shrinking factor $\gamma^t_l$. Therefore, our approach only requires simple calculations, eliminating the need for optimization. As reported in original Table 3, the 0.02-second increase corresponds to experiments with the ResNet20 model. To provide a broader perspective, we have also conducted experiments with other model architectures. The results are included in **Table 3** and are also summarized in the table below for your convenience. It can be observed that while larger models may result in a slightly higher computational load, our method remains significantly more efficient compared to optimization-based approaches. We have included this analysis and the results of these additional experiments in the revised manuscript to better illustrate the efficiency of FedLWS. Thank you for raising this point, which allowed us to provide a more comprehensive analysis.
>
> >[**Table:** Average aggregation execution time (Sec) across different model structures.]
> > | **Method**  | **CNN**  | **ResNet20** | **ViT**  | **WRN56_4** | **DenseNet121** |
> > | --- | -- | --- | ---- | -- | ---- |
> > | FedAvg   | 0.019     | 0.10   | 0.18   | 0.561   | 1.359   |
> > | FedLAW   | 4.830     | 7.11  | 9.80  | 20.08    | 27.25   |
> > | **FedLWS (Ours)** | **0.035** | **0.12**     | **0.21** | **0.832**   | **1.756**       |

---

> ### Author Response · Authors · 2024-11-24
> **Reply to Reviewer wdHM (Part 6)**
>
> **W10:** The authors should analyze under which experimental conditions FedLAW outperforms FedLWS and examine whether specific variables might explain this phenomenon.
>
> > **Response to W10:** Thank you for your insightful comment regarding the comparison between FedLWS and FedLAW. Following your feedback, we have provided a more in-depth analysis in lines [393-399] of the revised manuscript, summarized below for your convenience. It can be observed that FedLAW performs better on the CIFAR-10 dataset and, in some scenarios, even surpasses our method. One possible explanation is the proxy dataset used in our experiments, which contains 200 samples, consistent with the original settings in [a]. For CIFAR-10, this proxy dataset includes 20 samples per class, providing sufficient information to optimize the shrinking factor effectively. However, for datasets like CIFAR-100 and TinyImageNet, the proxy dataset contains only 2 and 1 sample per class, respectively. This limited representation makes it difficult for FedLAW to train an optimal shrinking factor, which may explain the variations in its performance across different scenarios.
> >
> > ---
> >
> > [a] Li, Zexi, et al. "Revisiting weighted aggregation in federated learning with neural networks." International Conference on Machine Learning. PMLR, 2023.
>
> **W11:** Sole text dataset, is insufficient to fully validate FedLWS’s performance on other types of text data.
>
> > **Response to W11:** We appreciate the reviewer’s suggestion to explore additional scenarios. In response, we conducted experiments on additional text datasets *Sogou News* and *Amazon Review* to further evaluate the robustness and generalizability of our approach. Notably, Amazon Reviews is a widely used dataset in Domain Adaptation. Due to its inherent heterogeneity in feature shifts, we utilized the dataset directly without applying Dirichlet partitioning. The results, as shown in the table below, align with our findings on the original dataset, demonstrating the effectiveness of our method across diverse scenarios.
> >
> > | **Method**  | **With LWS?** | **AG News** ($\alpha=0.1$) | **AG News** ($\alpha=0.5$) | **Sogou News** ($\alpha=0.1$) | **Sogou News** ($\alpha=0.5$) | **Amazon Review (Feature Shift)** |
> > | ----------- | ------------- | -------------------------- | -------------------------- | ----------------------------- | ----------------------------- | --------------------------------- |
> > | **FedAvg**  | ✗             | 73.43                      | 70.37                      | 87.68                         | 91.53                         | 88.15                             |
> > | **FedAvg**  | ✓             | **74.96**                  | **72.32**                  | **90.56**                     | **92.76**                     | **88.62**                         |
> > | **FedProx** | ✗             | 65.07                      | 74.56                      | 88.60                         | 92.28                         | 88.24                             |
> > | **FedProx**  | ✓             | **75.24**                  | **77.18**                  | **90.17**                     | **93.10**                     | **88.75**                         |
> >
> > These additional experiments reinforce our claim that the proposed method performs well across various scenarios and confirm its applicability beyond the initially considered dataset. Thank you for highlighting this important aspect.
>
> **W12:** Additional details on how FedLWS manages potential privacy risks would be useful. It would be valuable to discuss if FedLWS has any compatibility with existing privacy-preserving techniques.
>
> > **Response to W12:** Similar to foundational FL algorithms like FedAvg, our FedLWS only requires clients to transmit locally trained model parameters without sharing any additional information about their datasets. Therefore, our method does not introduce any additional privacy risks beyond those inherent in standard FL practices. Moreover, FedLWS is fully compatible with existing privacy-preserving techniques, such as differential privacy, as it only adds a simple weight shrinking step on the server side after model aggregation. This design ensures that FedLWS can seamlessly integrate with various FL frameworks and privacy-enhancing methods without requiring modifications to client-side operations or data transmission protocols.

---

> ### Author Response · Authors · 2024-11-24
> **Reply to Reviewer wdHM (Part 7)**
>
> **W13:** Adding a brief explanation of why FedLWS outperforms weight decay in the FL context would make this comparison more compelling.
>
> > **Response to W13:** Thank you for your insightful comment. As noted, AWD and AdaDecay were not originally designed for the federated learning (FL) context. In our work, we adapted these methods to make them applicable to FL settings for a fair comparison with FedLWS. However, these methods do not account for the differences between local models, which are critical in heterogeneous FL scenarios. In contrast, FedLWS explicitly addresses these differences by considering the variance of client update gradients, combined with its layer-wise adaptability, leading to better performance in FL environments.
>
> > We have included this explanation in **Appendix A.2** to clarify this point further. Thank you for your suggestion, which has helped improve the clarity of our work.
>
> **Q2:** Can the authors provide a detailed background on prior experimental findings to help readers understand the motivation behind the approach?
>
> > **Response to Q2:** The motivation for our approach stems from prior findings in federated learning that highlight the following challenges:
> >
> > - Weight shrinking is an intriguing phenomenon, demonstrating that when the sum of aggregation weights is less than 1, the model achieves better generalization. However, existing methods often rely on proxy datasets, which raise privacy concerns in federated learning scenarios and pose practical challenges for real-world applications.
> > - Significant variations exist between different layers of a model, and using a uniform shrinkage factor across all layers may adversely affect model aggregation.
> >
> > To address these challenges, we conducted the experiment shown in **Figure 1(b)**, which confirms that layer-wise weight shrinking can indeed further improve model performance. Additionally, to eliminate the need for proxy datasets, we explored the role of the shrinkage factor in the aggregation process and its ability to balance the relationship between regularization and optimization terms. We hypothesize that when client heterogeneity is high, stronger regularization should be applied during the aggregation process. To this end, we propose using the variance of global gradients as a measure of client heterogeneity. We validated this hypothesis through experiments in **Figures 1(a) and 7**, as well as the theoretical analysis presented in **Appendix D**.
>
> > These findings motivated us to design a layer-wise adjustment mechanism that explicitly accounts for heterogeneity while remaining computationally efficient and flexible. Our method dynamically calculates layer-wise shrinkage factors using the model's parameters, avoiding the need for additional optimization or proxy datasets.
>
> **Q3:** In the experimental section, why does the proposed method perform better, and can this be explained rather than just described?
>
> > **Response to Q3:** The improved performance of our method can be attributed to three main factors:
> >
> > - **Heterogeneity-Aware Design:** By leveraging client variance to calculate the shrinking factor, our adaptive approach dynamically adjusts the strength of the regularization according to the current heterogeneity between the clients during training, which reduces the generalization gap in FL models.
> > - **Layer-Wise Adaptability:** By computing shrinking factors specific to each layer, our method captures and mitigates inter-layer heterogeneity, leading to a more stable aggregation process.
> > - **Efficient Integration:** Our method introduces minimal computational overhead and is compatible with many existing FL methods, facilitating further enhancements in model performance building upon previous approaches.
> >
> > We have included additional explanations in the revised manuscript to better illustrate these points. In addition, based on the reviewers’ feedback, we have included a variance-based generalization bound in **Appendix D**, providing a theoretical foundation for managing client heterogeneity in federated learning (FL). The adaptive mechanism of our layer-wise shrinking approach dynamically adjusts the regularization strength based on client heterogeneity during training, thereby reducing the generalization gap. These findings underscore the effectiveness of our method in addressing client heterogeneity and achieving better model performance in FL.
>
> Overall, we hope that our responses can fully address your concerns and will be grateful for any feedback.

---

> > ### Comment · Reviewer_wdHM · 2024-11-30
> > **confirmed the score**
> >
> > Thank you for your thoughtful and detailed responses to each of the comments. The response made to the manuscript, including the new theoretical analyses and experimental results in both the main text and the appendix, have addressed the majority of the issues I raised previously. These additions significantly enhance the robustness and clarity of the paper. However, there are still some errors and unresolved questions that need attention:
> >
> > - In the experiments conducted on the newly added text datasets, there appear to be errors in the experimental table. Specifically, why are there three rows labeled as "FedAvg"? Based on the experimental context, the fourth row seems to correspond to experiments involving FedProx.
> >
> > - The inclusion of experiments involving CNN and ViT in Figure 9 is a valuable addition and helps extend the generalizability of FedLWS. However, I noticed that only the layer 20 of ResNet20 shows an unusual behavior where the γ value decreases initially and then increases—a pattern that appears counterintuitive. This phenomenon is not observed in other layers or architectures. Could the authors provide a clear explanation for this anomaly?
> >
> > - In the revised version, the authors modified some of the experimental results from the original manuscript, explaining that these changes were due to adjustments in the experimental settings. However, this raises an important question: why did adjusting the hyperparameters result in changes to only some of the results? For example, in Table 5, the results for FedLAW on CIFAR-10 differ from those in the original manuscript, while the results for CIFAR-100 remain unchanged. Were different parameter settings used for these datasets? If so, this should be explicitly clarified. The authors should provide a clear and detailed description of the experimental environment and specific hyperparameter settings in the manuscript.
> >
> > In conclusion, while the manuscript represents a substantial improvement and successfully resolves many prior concerns, the above issues still need to be addressed. I appreciate the authors' efforts in revising the manuscript and look forward to seeing these points clarified and corrected.

---

> ### Author Response · Authors · 2024-11-30
> **Further Reply to Reviewer wdHM**
>
> Thank you for your feedback! We greatly appreciate the time and effort you have invested in reviewing our manuscript and are glad to hear that the revised version has addressed many of the concerns raised. Below, we provide detailed responses to the remaining issues you mentioned:
>
> >**1. Error in the experimental table**: Thank you for pointing out the error in the table. This issue occurred during the conversion of the table from LaTeX to Markdown format. The experimental table for the text dataset (**Table 3**) is correct in the revised manuscript, and we have updated the table in the comments section. We truly appreciate your careful attention to detail and your valuable feedback.
>
>
> >**2. Unusual behavior in the $\gamma$ for layer 20 of ResNet20 in Figure 9:** Due to the differing scales of the x-axis in Figure 9, the comparison across different models may not be as intuitive and could potentially lead to some misunderstanding. To provide a clearer comparison of how $\gamma$ evolves across various model architectures and layers, we have redrawn the plots for the first 50 and 10 communication rounds for all models. These updated plots can be accessed via the following **anonymous link**: [https://anonymous.4open.science/r/FedLWS-A772/Figure9.pdf](https://anonymous.4open.science/r/FedLWS-A772/Figure9.pdf). We believe that this revision provides a clearer and more intuitive depiction of how the layer-wise $\gamma$ evolves during the training process. We will update **Figure 9** in the camera ready.
> >As for the behavior where $\gamma$ initially decreases and then increases, it is more like a fluctuation that can occur during training. Such fluctuations are more prominent in CNNs, which typically have fewer parameters and are therefore more sensitive to gradient updates. However, it is important to emphasize that the overall trend remains consistent across the models.
>
> >Additionally, a common characteristic observed across the three models in Figure 9 is that the final classifier layer (Layer 20 in ResNet, Layer 5 in CNN, and Layer 8 in ViT) shows more noticeable differences compared to the other layers. This aligns with previous research, which highlights the unique behavior of the classifier layer. This observation further supports the need for designing corresponding shrinking factors for each layer in our approach, enhancing its adaptability.
>
>
> >**3. Changes in experimental results due to hyperparameter adjustments:** We sincerely thank the reviewer for the insightful question. To ensure a fair comparison, we used the same learning rate of 0.08, weight decay of 5e-4, and the default hyperparameters recommended in the original papers for all methods. However, these default settings do not always yield optimal results across different experimental setups. While maintaining consistent hyperparameters across methods is important for a fair comparison, in order to demonstrate the effectiveness of our method, and following the reviewer’s suggestion, we adjusted the hyperparameters for the baseline methods that exhibited training instabilities in the original experiments (e.g., In Table 5, for FedLAW-CIFAR10-ResNet20, we set the local lr: 5e-3, server lr: 0.01, weight decay: 5e-5; for FedLAW-CIFAR100-WRN56_4, we set the local lr: 5e-3, server lr: 0.005, weight decay: 5e-5).
> >For scenarios where the baseline methods already demonstrated good performance, we did not make further adjustments to the hyperparameters.
> >We will provide a clear and detailed description of the experimental environment and specific hyperparameter settings in the camera ready to improve clarity.
>
> Overall, we hope these explanations clarify your concerns.

---

> > ### Comment · Reviewer_wdHM · 2024-11-30
> > **Reply to the Further Reply to Reviewer wdHM**
> >
> > Thank you for the clarification and confirmation. I believe the paper can be greatly improved if these points are included in the final version.

---

> > > ### Author Response · Authors · 2024-12-02
> > >
> > > Thank you for taking the time to review our response and for your valuable feedback.

---

### Official Review · Reviewer_2U5t · 2024-11-04

**Soundness:** 2
**Presentation:** 3
**Contribution:** 2
**Rating:** 6
**Confidence:** 3

**Summary:**

This work studies server-side aggregation algorithms for non-IID federated learning, proposing an adaptive layer-wise weight shrinking strategy. Compared to the existing weight shrinking method, the proposed approach does not require auxiliary datasets to adjust the weights and allows for different shrinkage across various layers of neural networks.

**Strengths:**

1. The layer-wise strategy is well-motivated, as non-IID data may have varying disparity effects on different layers/modules.

2. The proposal can be integrated into many non-IID-resistant training paradigms, such as FedProx and FedDisco.

**Weaknesses:**

1. The proposal presents similarities to existing concepts in federated learning, particularly the idea of layer-wise weighted aggregation (e.g., [a], [b]) and the weight shrinking technique from Li et al. (2023a). To enhance the clarity and impact of the proposed method, it would be beneficial for the authors to explicitly outline the novel aspects of their approach in comparison to these prior works.

[a] Lee, Sunwoo, Tuo Zhang, and A. Salman Avestimehr. "Layer-wise adaptive model aggregation for scalable federated learning." Proceedings of the AAAI Conference on Artificial Intelligence. Vol. 37. No. 7. 2023.

[b] Ma, Xiaosong, et al. "Layer-wise model aggregation for personalized federated learning." Proceedings of the IEEE/CVF Conference on Computer Vision and Pattern Recognition. 2022.

A comparison highlighting how the proposed method differs from or improves upon the layer-wise aggregation techniques discussed in [a] and [b] would be valuable. What unique contributions does this method bring to this area?

2. While the proposal shows strong empirical performance, it would be more persuasive with theoretical justification. Specifically, the authors might provide a theoretical analysis of the advantages (e.g., faster convergence rate) of layer-wise shrinking compared to the original weight shrinking approach from Li et al. (2023a).

**Questions:**

1. The relationship between the ratio of updating terms in Equation (4) and the gradient variance in Equation (5) is not immediately clear. It would be helpful if the authors could provide an intuitive explanation or theoretical evidence that supports this connection.

2. More guidance is needed on how to choose the hyperparameter $\beta$ and its potential impact on performance.

3. The current experiments only consider one text dataset; it would be beneficial to explore additional scenarios.

---

> ### Author Response · Authors · 2024-11-24
> **Reply To Reviewer 2U5t (Part 1)**
>
> **W1:** Clearly outline the novel aspects of this approach compared to prior works [a, b, c], emphasizing how it differs from or improves upon the layer-wise aggregation techniques in [a] and [b].
>
> > **Response to W1:**  Thank you for your insightful comment regarding the similarities between our work and existing approaches in federated learning. We appreciate the opportunity to outline the differences and novel contributions of our method compared to prior works.
> >
> > - **Comparison with Layer-Wise Aggregation Techniques ([a], [b]):** While our method shares the concept of layer-wise aggregation, there are key distinctions: The method in [a] adjusts the aggregation frequency for each layer to reduce communication costs while accounting for inter-layer differences. The method in [b] focuses on personalized FL, it designs a hyper-network to predict the layer-wise aggregation weights for each client. In contrast, our method is neither focused on aggregation frequency adjustment nor layer-wise aggregation weights adjustment. Instead, we propose an adaptive layer-wise weight shrinking step after model aggregation to mitigate aggregation bias, which is both computationally efficient and modular, enabling seamless integration with various FL frameworks and baselines.
> >
> > - **Comparison with weight shrinking technique [c]:** FedLAW [c] jointly learns the optimal shrinking factor $\gamma$ and aggregation weights $\lambda$ on the server; however, it relies on optimizing these factors using a proxy dataset, which is often impractical due to privacy concerns. Furthermore, FedLAW applies a uniform shrinking factor across all layers, overlooking inter-layer differences that are critical in non-IID scenarios. Our method, FedLWS, directly calculates the layer-wise shrinking factors using readily available gradients and global model parameters, eliminating the need for a proxy dataset. This approach also captures inter-layer differences, ensuring better adaptability to non-IID data, and significantly reduces computational overhead by avoiding server-side optimization. These innovations make FedLWS a flexible and efficient solution for federated learning.
> >
> > We have included this discussion in the revised manuscript to clearly outline the novel aspects of our method and its improvements over existing approaches. Thank you for this suggestion, which has helped us strengthen the clarity and impact of our work.
> >
> > ---
> >
> > [a] Lee, Sunwoo, Tuo Zhang, and A. Salman Avestimehr. "Layer-wise adaptive model aggregation for scalable federated learning." Proceedings of the AAAI Conference on Artificial Intelligence. Vol. 37. No. 7. 2023.
> >
> > [b] Ma, Xiaosong, et al. "Layer-wised model aggregation for personalized federated learning." Proceedings of the IEEE/CVF conference on computer vision and pattern recognition. 2022.
> >
> > [c] Li, Zexi, et al. "Revisiting weighted aggregation in federated learning with neural networks." International Conference on Machine Learning. PMLR, 2023.
>
>
>
> **W2&Q1:** Theoretical justification of the method and the connection between Equations (4) and (5).
>
> > **Response to W2&Q1:**  We appreciate the reviewer’s insightful questions and provide the following clarification:
> >
> > - Based on the reviewers’ feedback, we have included a variance-based generalization bound in **Appendix D**, providing a theoretical foundation for managing client heterogeneity in federated learning (FL). Specifically, when $\tau$ is large, it reflects high client gradient variance, indicating greater heterogeneity among clients. This leads to an increased generalization gap. The adaptive mechanism of our layer-wise shrinking approach dynamically adjusts the regularization strength based on client heterogeneity during training, thereby reducing the generalization gap.
> >
> > - Regarding the relationship between Equation (4) and Equation (5), the gradient variance $\tau$ reflects client heterogeneity, while the ratio determines the regularization strength via $\tau$. According to Equation (7) in the main paper, higher $\tau$ leads to lower $\gamma$, which strengthens the regularization. This dynamic adjustment mechanism leverages the gradient variance to guide the regularization strength, enhancing stability and reducing the generalization gap in the presence of high heterogeneity.
> >
> > In summary, the theoretical analysis provided in the appendix supports both the advantages of our layer-wise shrinking approach and the connection between Equations (4) and (5). These findings underscore the effectiveness of our method in addressing client heterogeneity and achieving better model performance in FL.

---

> ### Author Response · Authors · 2024-11-24
> **Reply To Reviewer 2U5t (Part 2)**
>
> **Q2:** More guidance is needed on how to choose the hyperparameter and its potential impact on performance.
>
> > **Response to Q2:**  Thank you for your valuable comment. We appreciate the opportunity to provide more guidance on how to choose the hyperparameter $\beta$ and its potential impact on performance. As shown in Eq.(7), $\beta$ controls the influence of $\tau$ on the shrinking factor $\gamma$, which directly affects the model’s adjustment process. Based on our experiments, we observed the following effects of $\beta$ on model performance: When $\beta$ approaches 0, $\gamma$ converges to 1, causing the model to degrade to the baseline. Conversely, when $\beta$ is too large, the calculated shrinking factor becomes excessively small, leading to model instability or failure. Our experiments indicate that a safe range for $\beta$ lies between 0.001 and 0.1.  In response to your comment, we have expanded the manuscript to include these observations and provided additional results in Figure 6 to illustrate the effects of $\beta$ over a broader range. This will help clarify the rationale for selecting $\beta$ and its impact on performance. Thank you again for highlighting this point, which has helped us improve the guidance provided in the paper.
>
> **Q3:** Only consider one text dataset; it would be beneficial to explore additional scenarios.
>
> > **Response to Q3:**  We appreciate the reviewer’s suggestion to explore additional scenarios. In response, we conducted experiments on additional text datasets Sogou News and Amazon Review to further evaluate the robustness and generalizability of our approach. Notably, Amazon Reviews is a widely used dataset in Domain Adaptation. Due to its inherent heterogeneity in feature shifts, we utilized the dataset directly without applying Dirichlet partitioning. The results, as presented in the table below, are consistent with our findings on the original dataset, thereby demonstrating the effectiveness of our method across different scenarios. These additional experiments strengthen our claim that the proposed method performs well under various scenarios and confirm its applicability beyond the initially considered dataset. Thank you for highlighting this important aspect.
> >
> > | **Method** | **With LWS?** | **AG News** (α = 0.1) | **AG News** (α = 0.5) | **Sogou News** (α = 0.1) | **Sogou News** (α = 0.5) | **Amazon Review** (Feature Shift) |
> > | ---------- | ------------- | --------------------- | --------------------- | ------------------------ | ------------------------ | --------------------------------- |
> > | FedAvg     | ×             | 73.43                 | 70.37                 | 87.68                    | 91.53                    | 88.15                             |
> > | FedAvg     | √             | **74.96**             | **72.32**             | **90.56**                | **92.76**                | **88.62**                         |
> > | FedProx    | ×             | 65.07                 | 74.56                 | 88.60                    | 92.28                    | 88.24                             |
> > | FedProx    | √             | **75.24**             | **77.18**             | **90.17**                | **93.10**                | **88.75**                         |
>
> Overall, we hope that our responses can fully address your concerns and will be grateful for any feedback.

---

### Author Response · Authors · 2024-11-24
**General Response**

We thank the reviewers for their time and effort in reviewing our manuscript. The valuable insights provided by the reviewers have enabled us to further improve the quality and readability of our manuscript. In response to the reviewers' questions and suggestions, we have made the following changes:

1. Provided a theoretical analysis of our method in **Appendix D**.
2. Discussed the relationship between our approach and existing work on layer-wise model aggregation in the Related Work section and clarified how our work differs.
3. Revised the relevant paragraphs in the Method section and corrected typos to improve the clarity and structure of the manuscript.
4. Adjusted the hyper-parameters of the baseline methods that did not converge or underperformed in the original experiments to ensure fairer comparisons.
5. Included more detailed analyses of experimental results in the Experiments section.
6. Updated the pseudocode to more intuitively illustrate how our method integrates with prior FL methods and to clearly show the additional computational steps required by our approach.
7. Conducted additional experiments and analyses based on reviewers' suggestions, which include:
   - Evaluation on more text datasets (**Table 2**).
   - Measuring algorithm execution times across a wider range of model architectures (**Table 3**).
   - Conducting hyperparameter experiments over a broader range (**Figure 6**).
   - Investigating the relationship between gradient variance $\tau$ and ratio $r$ across different scenarios (**Figure 7 in Appendix**).
   - Investigating the variations of layer-wise shrinking factors during training across a broader range of model architectures (**Figure 9 in anonymous link:** [https://anonymous.4open.science/r/FedLWS-A772/Figure9.pdf](https://anonymous.4open.science/r/FedLWS-A772/Figure9.pdf)).
   - Analyzing the effects on different layer types (**Table 12 in Appendix**).
   - Examining convergence trends during training (**Figure 10 in Appendix**).

We hope that these revisions address the reviewers' concerns and contribute to improving the rigor and comprehensiveness of our work.
We sincerely appreciate the reviewers' valuable feedback, which has significantly helped us refine our manuscript. Detailed responses to each reviewer’s specific comments are provided below.

---

### Comment · Area_Chair_BvUs · 2024-11-26
**Response**

Dear Reviewers,

The authors have provided their rebuttal to your questions/comments. It will be very helpful if you can take a look at their responses and provide any further comments/updated review, if you have not already done so.

Thanks!

---

### Meta-Review · Area_Chair_BvUs · 2024-12-20

**Metareview:**

This paper proposes a method of aggregation of local models in federated learning with a layer-wise adaptive shrinking factor every iteration. They show that it leads to improved test error.

This is an interesting idea with clear algorithmic bottomline. While the reviewers gave marginal scores, they are all positive about the paper.

I recommend acceptance.

**Additional Comments On Reviewer Discussion:**

The discussion phase was fruitful, most reviewers increased their scores.

---

### Decision · Program_Chairs · 2025-01-22

Accept (Poster)